# Circadian regulation of macromolecular complex turnover and proteome renewal

Estere Seinkmane [1], Anna Edmondson [1], Sew Y Peak-Chew [1], Aiwei Zeng [1], Nina M Rzechorzek [1], Nathan R James [1], James West [2], Jack Munns[1], David CS Wong [1], Andrew D Beale [1✉] & John S O'Neill [1✉]

## Abstract

**Although costly to maintain, protein homeostasis is indispensable for normal cellular function and long-term health. In mammalian cells and tissues, daily variation in global protein synthesis has been observed, but its utility and consequences for proteome integrity are not fully understood. Using several different pulse-labelling strategies, here we gain direct insight into the relationship between protein synthesis and abundance proteome-wide. We show that protein degradation varies in-phase with protein synthesis, facilitating rhythms in turnover rather than abundance. This results in daily consolidation of proteome renewal whilst minimising changes in composition. Coupled rhythms in synthesis and turnover are especially salient to the assembly of macromolecular protein complexes, particularly the ribosome, the most abundant species of complex in the cell. Daily turnover and proteasomal degradation rhythms render cells and mice more sensitive to proteotoxic stress at specific times of day, potentially contributing to daily rhythms in the efficacy of proteasomal inhibitors against cancer. Our findings suggest that circadian rhythms function to minimise the bioenergetic cost of protein homeostasis through temporal consolidation of protein turnover.**

**Keywords** Proteostasis; Circadian; SILAC; Ribosome
**Subject Categories** Post-translational Modifications & Proteolysis; RNA Biology; Translation & Protein Quality

## Introduction

Protein homeostasis, or proteostasis, refers to the dynamic process of maintaining protein abundance and functionality. It involves the regulation of synthesis, folding, localisation and degradation of proteins, such that the appropriate proteins are present within the appropriate concentration range, in the correct compartment, at the right time. Multiple quality control and stress response mechanisms function to preserve proteome integrity over multiple timescales (Wolff et al, 2014; Harper and Bennett, 2016), whereas failure of proteostasis networks is strongly associated with impairment of cell function as well as ageing-related pathological states such as neurodegeneration (Labbadia and Morimoto, 2015; Hipp et al, 2019). By contrast, priming of proteostatic pathways enhances cellular resistance to proteotoxic stress (Rzechorzek et al, 2015).

Most aspects of mammalian cellular and organismal physiology are regulated over a circadian (about daily) timescale to anticipate the differing demands of day and night (Dibner et al, 2010; Atger et al, 2017). Whilst circadian rhythms are generated cell autonomously (Welsh et al, 2004; Yoo et al, 2004), in vivo, myriad cellular clocks throughout the body are synchronised with daily environmental cycles by systemic timing cues. For example, daily rhythms of feeding entrain cellular clocks through the insulin signalling pathway to stimulate PERIOD"clock protein" production via activation of mammalian target of rapamycin complexes (mTORCs) (Crosby et al, 2019). Daily rhythms of PERIOD and mTORC activity facilitate daily rhythms of gene expression and protein synthesis. In particular, mTORC1 is a master regulator of bulk 5'-cap-dependent protein synthesis, degradation and ribosome biogenesis (Valvezan and Manning, 2019) whose activity is circadian-regulated in tissues and in cultured cells (Ramanathan et al, 2018; Feeney et al, 2016a; Stangherlin et al, 2021b; Mauvoisin et al, 2014; Jouffe et al, 2013; Sinturel et al, 2017; Cao, 2018). It is plausible that daily rhythms of mTORC activity underlie many aspects of daily physiology (Crosby et al, 2019; Stangherlin et al, 2021a; Beale et al, 2023b).

Most models for circadian regulation of mammalian cell function have suggested that daily rhythms in the transcription of 'clock-controlled genes' leads to daily rhythms in the abundance, and thus activity, of the encoded protein (Cox and Takahashi, 2019; Zhang et al, 2014; Andreani et al, 2015). However, recent -omics approaches, which measure many thousands of individual transcripts and proteins, have revealed multiple discrepancies with this abundance-based hypothesis, such as poor correlations between mRNA and encoded protein abundance (Stangherlin et al, 2021a). Moreover, the rather modest extent of daily changes in protein abundance (typically <20%), and poor reproducibility between independent studies (Janich et al, 2015; Mauvoisin et al, 2014; Reddy et al, 2006; Robles et al, 2014; Mauvoisin and Gachon, 2020;

[1]MRC Laboratory of Molecular Biology, Francis Crick Avenue, Cambridge CB2 0QH, UK. [2]Department of Medicine, University of Cambridge, Cambridge, UK.
✉E-mail: abeale@mrc-lmb.cam.ac.uk; oneillj@mrc-lmb.cam.ac.uk

Brooks et al, 2023), suggests that physiological variation in protein abundance is unlikely to account for large daily variations in multiple biological functions observed in tissues and cultured cells. Indeed, daily cycles of protein abundance would appear contrary to the essential requirement for maintaining proteostasis, which the major fraction of cellular energy budgets are spent to sustain (Buttgereit and Brand, 1995; Lane and Martin, 2010).

Compelling evidence for physiological daily variation in global rates of protein synthesis cannot be ignored, however (Lipton et al, 2015; Feeney et al, 2016a; Stangherlin et al, 2021b). Such observations are difficult to reconcile with linked observations that, excepting feeding-driven changes in mouse liver, total cellular volume and protein levels show little daily variation (Stangherlin et al, 2021b; Hoyle et al, 2017; Sinturel et al, 2017). To resolve these apparent discrepancies we have proposed that, in non-proliferating cells, daily changes in protein synthesis are accompanied by changes in protein degradation (Stangherlin et al, 2021a), resulting in daily cycles of protein turnover. This further predicts that daily rhythms in protein turnover prevail over rhythms in protein abundance to favour rhythmic proteome renewal over compositional variation. Daily turnover rhythms would be particularly beneficial for coordinated biogenesis of multiprotein complexes, since complex assembly requires individual subunits to be present stoichiometrically at the same time, in the same cellular compartment, or else be wastefully degraded (Juszkiewicz and Hegde, 2018; Taggart et al, 2020).

Here, we aimed to test these predictions by investigating circadian regulation of global protein synthesis and degradation, as well as macromolecular complex turnover, specifically. In so doing, we utilised bulk pulse-chase labelling to establish proof-of-principle, before developing a novel time- and cellular fraction-resolved dynamic mass spectrometry approach that provides the first direct and simultaneous measurements of protein synthesis and abundance proteome-wide. To validate our findings, we then tested for rhythmic macromolecular complex turnover directly by quantification of nascent ribosome complex assembly through the combination of heavy uridine pulse-labelling of nascent RNA with ribosome purification. Given its importance in health and disease, we also aimed to investigate the functional consequences of rhythmic proteostasis regulation, revealing time-of-day-dependent differential sensitivity to proteotoxic stress in both cells and mice.

# Results

## Phase-coherent global rhythms in protein synthesis and degradation

We first asked whether there was any indication of cell-autonomous daily variation in protein turnover in confluent cultures of non-transformed quiescent lung fibroblasts derived from mice expressing the rhythmic luciferase reporter PER2::LUC. Using this cellular model under constant conditions, longitudinal bioluminescence recordings from parallel replicate cultures can be used to provide a robust report of cell-autonomous circadian timekeeping and establish circadian phase (Yoo et al, 2004; Feeney et al, 2016b). To measure protein degradation in parallel with synthesis, and thus assess the level of protein turnover, we first

employed a traditional $^{35}$S-methionine/cysteine pulse-chase labelling strategy (15-min pulse, 60-min chase).

The experiment was performed over a 24-h time series followed by soluble protein extraction using digitonin, which preferentially permeabilises the plasma membrane over organelle membranes. For pulse alone, $^{35}$S incorporation varied significantly over this period (Fig. 1A,B; Appendix Fig. S1A), consistent with previous reports of rhythmic protein synthesis (Stangherlin et al, 2021b; Lipton et al, 2015; Zhuang et al, 2023). As expected, ~20% of nascently synthesised proteins had been degraded after 1 h of chase, representing rapid quality control-associated proteasomal degradation of orphan subunits as well as aberrant translation products due to premature termination and/or protein misfolding (Schubert et al, 2000; Wheatley et al, 1980; Harper and Bennett, 2016). Importantly, the proportion of degraded protein varied over time, being highest at around the same time as increased protein synthesis (Fig. 1B), indicating time-of-day variation in digitonin-soluble protein turnover which cannot be solely attributed to previously reported circadian regulation of protein solubility (Stangherlin et al, 2021b). Rather, it suggests that global rates of protein degradation may be coordinated with protein synthesis rates, and may vary over the circadian cycle.

Quality control-associated degradation predominantly occurs via the ubiquitin–proteasome system (UPS) (Schubert et al, 2000; Wang et al, 2013). To directly test whether the global rate of proteasomal protein degradation is under circadian control, as suggested previously (Desvergne et al, 2016; Ryzhikov et al, 2019; Hansen et al, 2021), we employed biochemical in-cell assays for proteasomal activity at discrete biological times over the circadian cycle. Over 2 days under constant conditions, we observed a significant ~24 h oscillation in proteasomal trypsin-like and chymotrypsin-like activities of the proteasome, but not caspase-like activity (Fig. 1C). Moreover, we detected a significant interaction between genotype and biological time when comparing trypsin-like proteasome activity between wild-type and Cryptochrome1/2-deficient cells, that lack canonical circadian transcriptional feedback repression (Appendix Fig. S1B; Wong et al, 2022). Previous proteomics studies under similar conditions have revealed minimal circadian variation in proteasome subunit abundance (Wong et al, 2022), suggesting that proteasome activity rhythmicity, and therefore rhythms in UPS-mediated protein degradation, are regulated post-translationally (Marshall and Vierstra, 2019; Hansen et al, 2021).

If global rates of proteasome-mediated protein degradation vary in phase with protein synthesis over the circadian cycle, this would result in circadian organisation of nascent protein turnover. To validate this, we employed puromycin, an antibiotic which is incorporated into nascent polypeptides by both elongating and stalled ribosomes (Nathans, 1964; Semenkov et al, 1992) thus making these peptides amenable to immunodetection (Aviner, 2020; Goodman and Hornberger, 2013; Schmidt et al, 2009). Unlike $^{35}$S-labelled and other amino acid analogues, puromycin can be added to cell media directly, without the need to remove endogenous amino acids, thereby minimising acute perturbations. Whilst most studies have used puromycin incorporation as a proxy for translation rate, we reasoned that protein degradation should also affect the observed levels of puromycylated peptides, as these peptides are prematurely terminated and are thus identified and degraded rapidly by the ubiquitin–proteasome system (Lacsina

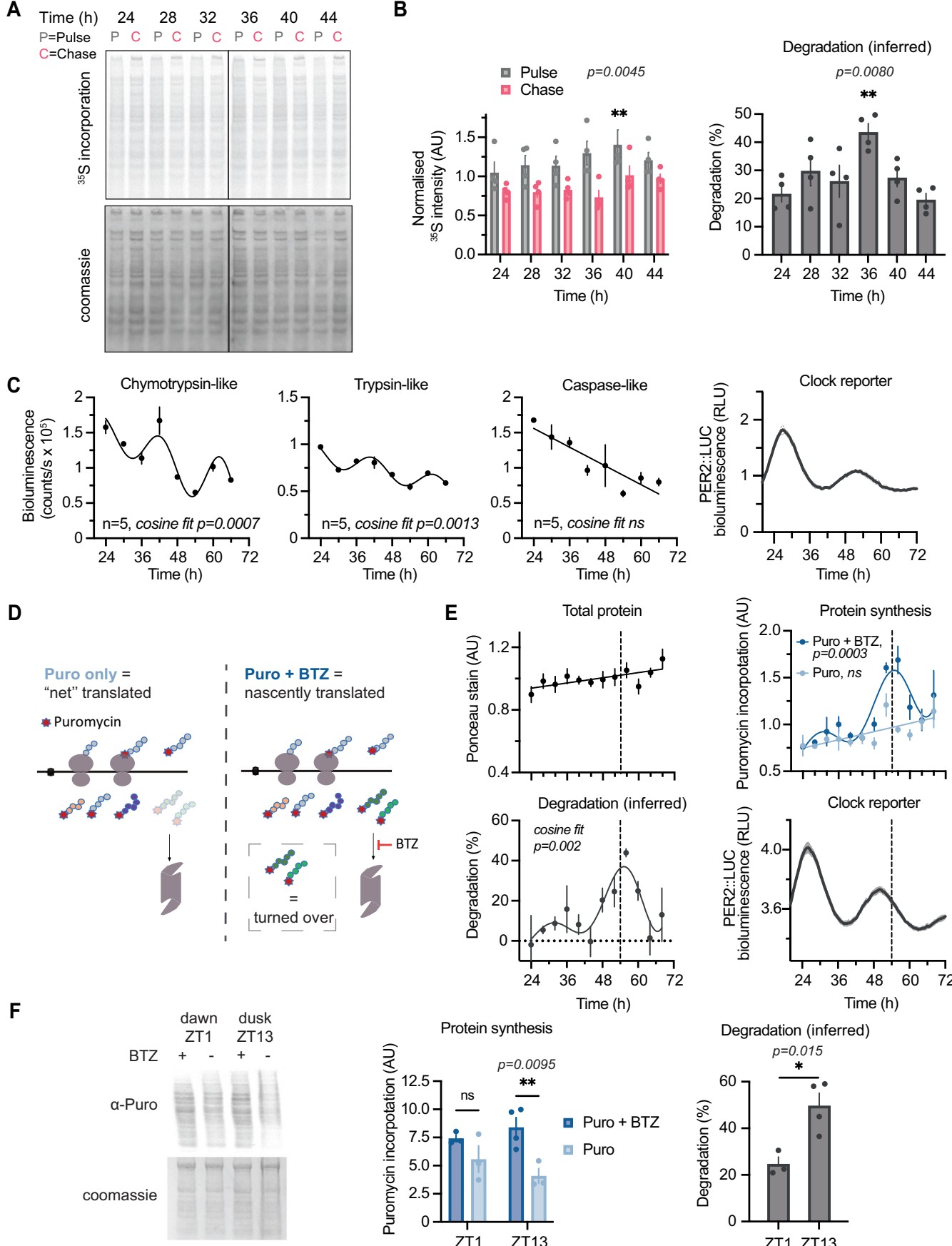

**Figure 1.   Bulk protein synthesis and degradation show circadian variation in culture and day/night differences in vivo.**

(A) A representative phosphor screen exposure of SDS-PAGE gel showing $^{35}$S-Met/Cys incorporation in 15 min pulse (P) and 1 h chase (C) samples at different circadian times in mouse lung fibroblasts lysed with digitonin buffer. (B) (Left) Quantification of radiolabel signal in pulse and chase at the different timepoints, normalised to protein content (by Coomassie stain) in four replicates shown in Appendix Fig. S1A. Statistics: two-way ANOVA with Dunnett's multiple comparison test comparing T24 to other timepoints shown, $P = 0.0080$. (Right) The inferred degradation within 1 h of chase for each replicate is plotted, calculated as 100%*(1-Chase/Pulse) for each replicate. Statistics: one-way ANOVA with Dunnett's multiple comparison test comparing T24 to other timepoints shown, $P = 0.0045$. $n = 4$ replicate cell cultures. (C) Chymotrypsin-like, trypsin-like, and caspase-like proteasome activities, measured by ProteasomeGlo cell-based assays, at different circadian times as indicated. Statistics: damped cosine wave fit compared with straight line (null hypothesis) by extra sum-of-squares F test, the statistically preferred fit is plotted and $P$ value displayed. Parallel PER2::LUC bioluminescence recording is shown on the right, acting as phase marker. $n = 5$ replicate cell cultures. (D) Schematic representation of the optimised puromycin incorporation assay. Puromycin (Puro) is incorporated into nascent peptide chains during translation elongation. A subset of these gets degraded by the proteasome within the 30 min labelling timeframe, resulting in a measure of "net" translation. In the presence of bortezomib (BTZ), the proteasome is inhibited, so the peptides that would have been degraded are still present and can be detected. Thus, all nascently translated peptide chains can be detected. Degradation can be inferred from comparing the two conditions, and allows estimation of both translation and degradation in a single assay. (E) Quantification of total protein, protein synthesis and protein turnover from a puromycin incorporation timecourse, where at each of the 12 timepoints Puro ± BTZ was added directly to cell media, and cells lysed 30 min afterwards in urea/thiourea buffer. Puromycin incorporation was assessed by Western blotting, and total protein from a parallel Ponceau Red stain. Degradation was inferred from Puro ± BTZ intensity, calculated as 100%*(1-Puro/Puro+BTZ). Statistics: damped cosine wave fit compared with a straight line (null hypothesis) by extra sum-of-squares F test, the statistically preferred fit is plotted and $P$ value displayed. Parallel PER2::LUC bioluminescence recording is shown below right, acting as phase marker (example phase comparison across all measurements is shown as dashed line). $n = 3$ replicate cell cultures. (F) Puromycin incorporation in vivo: mice received an i.p. injection of puromycin with or without BTZ at ZT1 or ZT13 (note: zeitgeber = 'time-giver' and indicates hours since initial lights on). Livers were harvested 40 min afterwards and extracted with urea/thiourea buffer. Representative anti-puromycin Western blot is shown (left), and quantification is shown (centre), normalised for protein loading as assessed by Coomassie staining. (Right) The inferred degradation is plotted, calculated as 100%*(1-Puro/Puro+BTZ). Statistics: two-way ANOVA with Sidak's multiple comparisons test (centre), unpaired $t$ test (right). $N = 4$ mice were used per condition, but in some cases one of the four injections were not successful, i.e., no puromycin labelling was observed and so no quantification could be performed (full data in Appendix Fig. S2B). Data information: In (B, C, E, F), data are presented as mean ± SEM. *$P \leq 0.05$, **$P < 0.01$. Source data are available online for this figure.

et al, 2012; Szeto et al, 2006). Acute (30 min) puromycin treatment of cells in culture, with or without proteasomal inhibition (by bortezomib, BTZ), allowed us to measure both total nascent polypeptide production (+ BTZ) and the number of nascent polypeptides remaining when the UPS remained active (−BTZ). This allowed inference of the level of UPS-mediated degradation of puromycylated peptides within each time window, as a proxy for nascent protein turnover (Fig. 1D).

Over 2 days under constant conditions, puromycin incorporation in the presence of BTZ, i.e., total nascent polypeptide production, showed significant circadian variation. In contrast, cells that were treated with puromycin alone showed no such variation, and nor did total cellular protein levels (Fig. 1E; Appendix Fig. S2A). These observations support the possibility that phase-coherent daily rhythms in protein degradation might act in parallel with rhythms in translation rate, such that the proportion of degraded peptides vary in synchrony with those that were translated, resulting in a daily variation of protein turnover without variation in protein abundance (Fig. 1E).

Protein synthesis is the most energetically expensive process that most cells undertake (Buttgereit and Brand, 1995; Lane and Martin, 2010). In vivo, the temporal consolidation of global translation might be expected to confer a fitness advantage by organising this energetically expensive process to coincide with the biological time of greatest (anticipated) nutrient availability. In nocturnal mice, for example, hepatic ribosome biogenesis preferentially occurs at night, during the active/feeding phase (Jouffe et al, 2013; Jang et al, 2015; Sinturel et al, 2017). To explore the physiological relevance of our cellular observations, we adapted the puromycin ± BTZ labelling strategy in vivo, to test the specific prediction that nascent protein turnover is increased during the active phase compared with the rest phase. In livers isolated from mice at opposite times of day, we observed significantly higher turnover during the night (active/feeding phase, ZT13), compared with the daytime (rest/fasting phase, ZT1) (Fig. 1F; Appendix Fig. S2B). We suggest that both in cells and in mouse liver in vivo, the daily variation in nascent

protein turnover could be an expected consequence of imperfect translation, which requires ubiquitous protein quality control mechanisms to shield the proteome from defective, misfolded, or orphaned (excess subunit) proteins.

## Proteome-wide investigation of circadian protein synthesis, abundance, and turnover

Beyond protein quality control, Fig. 1 invited us to consider how circadian regulation of global protein degradation might interact with rhythmic synthesis to impact proteome composition more broadly, at the level of individual proteins. In the simplest scenario, protein abundance would correlate with synthesis rate; however, rhythmic degradation might attenuate variation in the abundance of rhythmically synthesised protein, or alternatively it may generate variation in the abundance of constitutively synthesised proteins. We devised a novel proteome-wide approach to test each of these scenarios.

To measure protein production and abundance simultaneously and directly over the circadian cycle, we utilised pulsed stable isotopic labelling with amino acids in culture (pSILAC), in combination with TMT-based mass spectrometry quantification to allow multiplexed measurements. Although pSILAC is normally applied continuously and protein harvested at multiple points to measure half-life (Doherty et al, 2009; Schwanhäusser et al, 2011; Ross et al, 2021), here we used a repeated fixed time window for SILAC labelling to measure newly synthesised proteins (Fig. 2A,B). To enable sufficient heavy labelling for detection, a 6-h time window was employed, thus measuring synthesis and abundance within each quarter of the circadian cycle.

Over our 48-h time series, we reliably detected heavy peptides for 2528 unique proteins from whole-cell lysates, representing estimates of their synthesis at each time window, and compared this with their total abundance calculated from the sum of heavy and light peptides (Dataset EV1; examples in Fig. 2C). The specific number or proportion of rhythmically synthesised and/or abundant

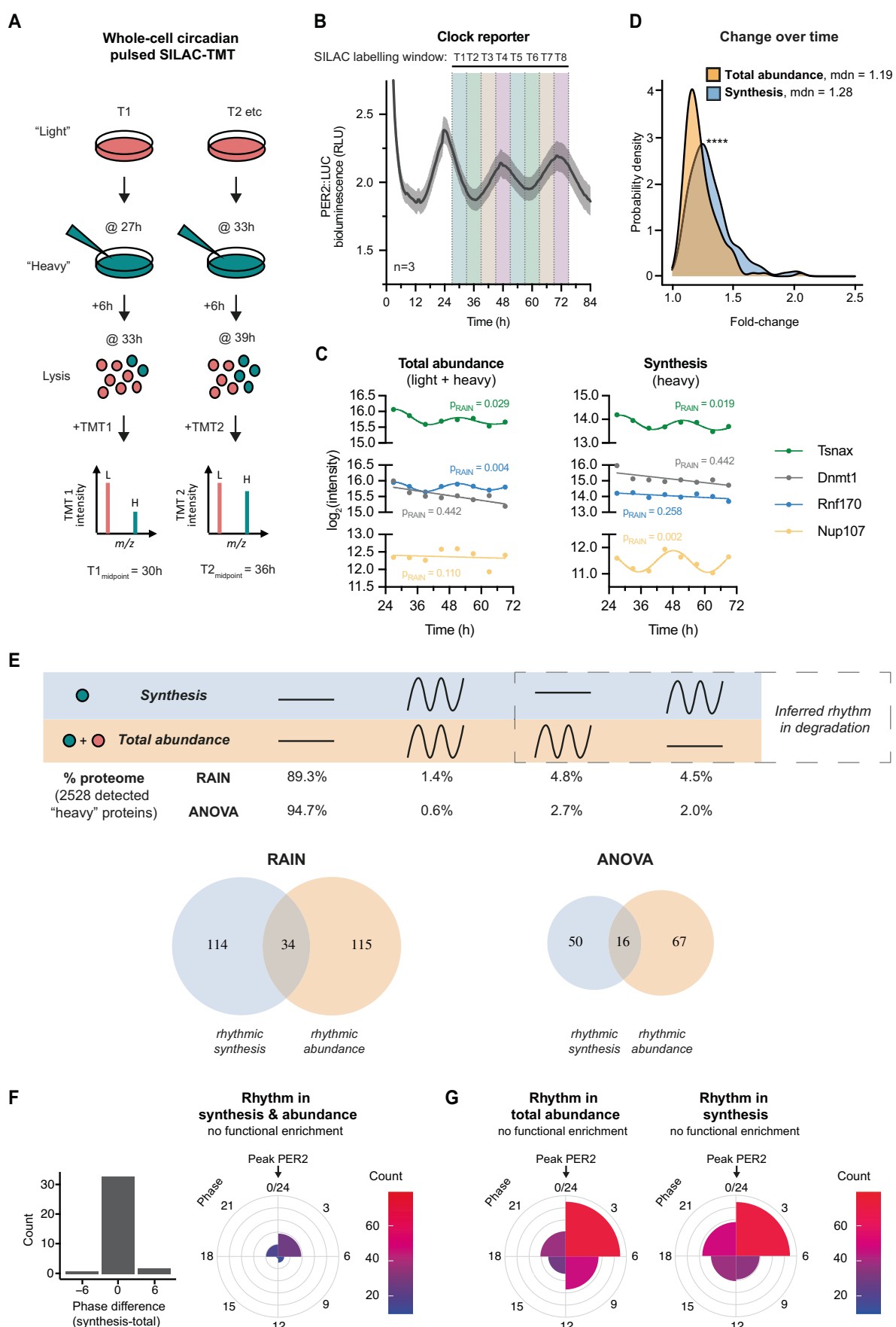

**A** Whole-cell circadian pulsed SILAC-TMT

**B** Clock reporter

**C** Total abundance (light + heavy) / Synthesis (heavy)

**D** Change over time

**E**

**F** Rhythm in synthesis & abundance — no functional enrichment

**G** Rhythm in total abundance — no functional enrichment / Rhythm in synthesis — no functional enrichment

◄ **Figure 2.  Proteome-wide investigation of circadian protein synthesis, abundance, and turnover using pulsed-SILAC mass spectrometry.**

(A) Schematic of circadian pulsed SILAC-TMT experiment design. In a set of entrained fibroblasts, at each circadian timepoint "light" media (DMEM with standard L-Arg and L-Lys) is switched for "heavy" (DMEM with $^{13}C_6{}^{15}N_4$ L-Arg and $^{13}C_6{}^{15}N_2$ L-Lys), and cells lysed after 6 h. Lysates are then digested, labelled with tandem mass tags (TMT), mixed and analysed by mass spectrometry. Data are presented aligned with the midpoint of the labelling window. (B) Parallel bioluminescence recording of PER2::LUC, acting as phase marker, overlayed with SILAC labelling windows that were used for the timecourse. (C) Representative examples of proteins (taken from Dataset EV1) changing rhythmically or staying constant in their total abundance (left) or synthesis (right), as measured in the pSILAC-TMT timecourse and rhythmicity defined by the RAIN rhythm detection method at $P < 0.05$. Representative of rhythmic abundance and synthesis: Akap12; representative of arrhythmic abundance and synthesis: Dnmt1; representative of rhythmic abundance but arrhythmic synthesis: Rnf170; and representative of arrhythmic abundance but rhythmic synthesis: Nup107. (D) Probability density distribution of fold change between peak and trough for proteins rhythmic in synthesis and in total abundance, representing the extent of change over time in these two sets. Statistics: Mann–Whitney test, $P < 0.0001$. (E) Comparison of rhythmicity between individual proteins' synthesis and total abundance. Statistically significant change over time was assessed by two algorithms, RAIN and ANOVA, with $P < 0.05$ taken as rhythmicity threshold. (Top) Percentages of detected proteins falling under the four rhythmicity categories by the two algorithms are displayed. (Bottom) Venn diagrams of proteins significant for rhythms in synthesis and/or abundance. Degradation rhythms can account for cases of proteins with rhythms in synthesis but not abundance, or vice versa. Overall, 6264 proteins were detected; out of those at least one heavy peptide was detected for 2528 proteins (the set used for the analysis). (F) Phase distribution of proteins rhythmic (with RAIN threshold of $P < 0.05$) in both synthesis and total abundance, as well as their circadian phase difference. Gene ontology functional enrichment was tested for by GOrilla tool, in each phase separately or together, against the background of all detected proteins, and no terms were significant below the corrected $P$ value (FDR $q$-value) 0.05 cutoff. Phase 0 is set as the peak of PER2::LUC. (G) Phase distribution of proteins rhythmic (with RAIN threshold of $P < 0.05$) in their total abundance and in their synthesis. Phase 0 is set as the peak of PER2::LUC. Data information: In (B), data are presented as mean ± SEM. ****$P < 0.0001$. Source data are available online for this figure.

proteins is expected to vary with detection method (Hughes et al, 2017; Mei et al, 2021) and may be susceptible to overestimation of rhythmicity. We therefore employed several methods, including less stringent RAIN and more stringent ANOVA, to compare the extent of temporal variation in protein synthesis and total abundance (Fig. 2D,E).

Consistent with similar previous studies, <10% of detected proteins showed any significant variation in abundance over the circadian cycle (Fig. 2E). This is also expected considering the long average half-life (~days) of mammalian proteins (Mathieson et al, 2018; Schwanhäusser et al, 2011; Wong et al, 2022). Of the rhythmically abundant proteins, a minority showed accompanying rhythms in synthesis, with no difference in phase (Fig. 2E,F). The proportion of such proteins was more than expected by chance ($P < 0.0001$, Fisher's Exact Test), and their behaviour aligns with the canonical "clock-controlled gene" paradigm, in which physiological rhythms are proposed to arise through circadian variation in protein abundance, generated via transcriptional and translational oscillations. Strikingly however, the majority of rhythmically synthesised proteins showed no accompanying rhythm in abundance and vice versa (Fig. 2E, right), and therefore must have an aligned rhythm in degradation. Similar proportions of the proteome showed rhythms in synthesis as rhythms in abundance (Fig. 2E). However, the extent of daily synthesis variation (fold change) was significantly greater than abundance (Fig. 2D). These observations are consistent with our model of widespread temporal organisation of protein degradation within the circadian-regulated proteome.

Considering all detected proteins that were either rhythmically synthesised or rhythmically abundant, peaks occurred in all four quarters of the cycle (Fig. 2G), but clustered in the quarter of the cycle immediately after the peak of PER2::LUC, in accordance with proteome-wide rhythms (Fig. 1A,B). Gene ontology analysis did not reveal functional enrichment for any particular biological process or compartment in either group compared with the background.

## Targeted investigation of circadian protein synthesis, abundance, and turnover

The experiment above was designed to combine and compare two time-resolved processes—that of circadian variation and that of protein production—and so only considered proteins with reliably

detectable heavy label incorporation within a given labelling window (6 h) across all timepoints. This inevitably limited and biased the proteome coverage towards abundant proteins with higher synthesis rates, irrespective of cellular compartment or function. This probably explains the absence of functional enrichment among rhythmic proteins that have been observed in other studies, as well as the lower level of overall variation in synthesis than would be expected from the bulk labelling investigations.

To gain more insight into the dynamics of circadian proteomic flux, we refined our pSILAC approach, this time focusing on proteins in complexes. Specifically, we aimed to test the hypothesis that circadian control of translation and turnover facilitates the coordinated assembly of multiprotein complexes (Taggart et al, 2020; O'Neill et al, 2020; Stangherlin et al, 2021a). To achieve this, we utilised and adjusted LOPIT-DC protocol (Geladaki et al, 2019) which was developed to separate different cellular compartments and fractions, to isolate the macromolecular complex (MMC) fraction via gentle cell lysis followed by sequential ultracentrifugation.

To further improve the sensitivity of our pSILAC method to circadian differences, especially those occurring among the most recently synthesised proteins, we also employed a shorter pulse (1.5 h). To compensate for the shorter pulse and the fractionation, which otherwise would have resulted in decreased coverage, especially of the heavy peptides, we employed further technical improvements. Namely, we added a so-called booster channel: an additional fully heavy-labelled cell sample within a TMT mixture (Klann et al, 2020). When the mixture is analysed by MS, heavy peptides from the booster channel increase the overall signal of all identical heavy peptides at MS1 level; at MS2 and MS3, this results in improved detection of heavy proteins in the other TMT channels of interest and is particularly advantageous for the proteins with the lower turnover that would fall below the MS1 detection limit without the booster.

With this new design, heavy peptides within the enriched MMC fraction were quantified across 2 days (as for the first pSILAC experiment), representing proteins synthesised within 1.5 h at each timepoint (Fig. 3A,B; Dataset EV2). Despite enriching for only one cellular compartment, the overall coverage in this experiment was similar to the previous one (6577 and 6264 proteins, respectively), due to the altered and more targeted approach; with heavy peptides

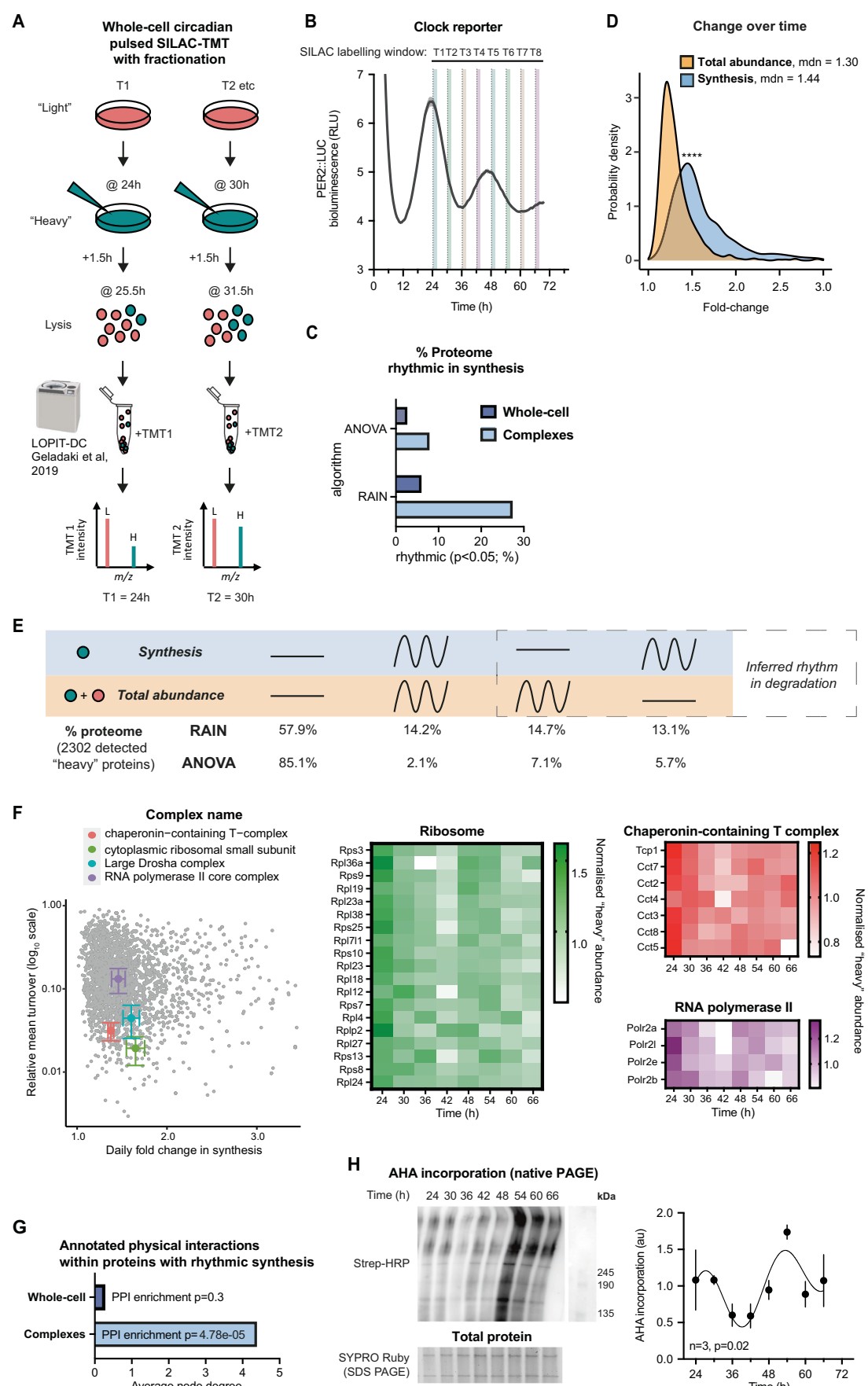

◀ **Figure 3. Targeted analysis of circadian complex biogenesis, abundance, and turnover by fractionated pulsed-SILAC mass spectrometry.**

(A) Schematic of circadian pulsed SILAC-TMT with extra fractionation step. In a set of entrained fibroblasts, at each circadian timepoint "light" media (DMEM with standard L-Arg and L-Lys) is switched for "heavy" (DMEM with $^{13}C_6^{15}N_4$ L-Arg and $^{13}C_6^{15}N_2$ L-Lys), and cells collected after 1.5 h. Samples were then subjected to sequential ultracentrifugation, a method adjusted from Geladaki et al, 2019 LOPIT-DC protocol. The fraction enriched for macromolecular protein complexes (MMC fraction) was labelled with tandem mass tags (TMT), mixed and analysed by mass spectrometry. $N = 2$ replicate cell cultures per timepoint. (B) Parallel bioluminescence recording of PER2::LUC, acting as phase marker, overlayed with SILAC labelling windows that were used for the timecourse. Normalised mean ± SEM are shown as line and shading, respectively. $n = 4$ replicate cell cultures. (C) Comparison of percentage of detected proteins that are considered rhythmic ($P < 0.05$) in their synthesis by the two algorithms used, between the whole-cell pSILAC experiment, presented in Fig. 2, and the experiment focused on complexes, presented in this figure. (D) Probability density distribution of fold change between peak and trough for proteins rhythmic in synthesis and in total abundance, representing the extent of change over time in these two sets. Statistics: Mann–Whitney test. (E) Comparison of rhythmicity between individual proteins' synthesis and total abundance. Statistically, significant change over time was assessed by two algorithms, RAIN and ANOVA, with $P < 0.05$ taken as rhythmicity threshold. Percentages of detected proteins falling under the four rhythmicity categories by the two algorithms are displayed. Overall, 6577 proteins were detected; out of those at least one heavy peptide was detected for 2302 proteins (the set used for the analysis). (F) Coordinated turnover of proteins belonging to complexes: for four selected complexes, their detected annotated subunits (according to a compilation of CORUM, COMPLEAT and manual annotations) were averaged in terms of their fold change over time (x axis), and relative turnover (proportion of heavy to total peptide intensity averaged across 8 timepoints, y axis). Chaperonin-containing T-complex, $n = 7$; cytoplasmic ribosomal small subunit, $n = 17$; large Drosha complex, $n = 14$; RNA polymerase II complex, $n = 5$. All proteins are displayed in the background in grey. The normalised heavy abundance of these proteins over time, representing new synthesis, is shown on heatmaps on the right. (G) Proteins rhythmic (RAIN $P < 0.05$) in their synthesis were analysed using STRING interaction database, filtering for high-confidence, physical interactions. Proteins with rhythmic synthesis in the complex fraction had an interconnected protein–protein interaction network, with high average node degree and significant enrichment in interactions over all detected proteins in that experiment, whereas for proteins with rhythmic synthesis in the whole-cell experiment (Fig. 2) this was not the case. (H) Fibroblasts were pulsed with AHA for 1.5 h at the indicated timepoints, and AHA incorporation into protein complexes and other higher molecular weight species was measured, using biotin as click substrate and streptavidin-HRP for detection after non-denaturing gel electrophoresis (left, top). Signal was quantified (right) after normalisation to total protein content as measured by SYPRO Ruby on a parallel SDS-PAGE gel (left, bottom). Statistics: damped cosine wave fit (plotted) preferred over straight line, extra sum-of-squares F test P value displayed. $n = 3$ replicate cell cultures. Data information: In (B, H), data are presented as mean ± SEM of replicate cell cultures. In (F), data are presented as mean ± SEM of values for members of each indicated complex. *$P \leq 0.05$, ****$P < 0.0001$. Source data are available online for this figure.

detected for 2302 proteins. There was a significant circadian variation among the overall amount of heavy-labelled peptides that was consistent with the rhythmic production of nascent proteins, whereas the total protein level in this fraction showed no change over time (Appendix Fig. S3A,B).

Using boosted fractionated pSILAC, we immediately noticed a threefold increase in the proportion of proteins that varied significantly over time in their synthesis as compared to the whole-cell level, regardless of the algorithm used (Fig. 3C). The production of rhythmically synthesised proteins in this fraction also varied over time to a far greater extent than did their abundance (~twofold greater variation, Fig. 3D). Moreover, we found that a much higher proportion of detected proteins exhibited rhythms in both synthesis and total abundance than was observed at the whole-cell level (Fig. 3E). As in whole-cell, the proportion of proteins showing rhythmic synthesis but not abundance and vice versa, was much greater than expected by chance ($P < 0.0001$, Fisher's Exact Test). By inference, therefore, the proportion of proteins that are rhythmically degraded in this fraction must equal or exceed the proportion that are rhythmically synthesised. It is also noteworthy that although there were small sets of proteins that were rhythmic in both whole-cell (Fig. 2) and MMC fractions (Fig. 3), in both synthesis and total abundance, none of these four overlaps were higher than would have been expected by chance.

Analysis of the proteins in the MMC fraction revealed 243 annotated multiprotein complexes (from CORUM, COMPLEAT and manual annotation (Ori et al, 2016; Giurgiu et al, 2019)) to be present, including 82 complexes for which half or more annotated subunits were detected (Dataset EV3). It has previously been shown that protein subunits within the same complex tend to share similar turnover rates, which is thought to facilitate their coordinated assembly and removal (Price et al, 2010; Mathieson et al, 2018). We observed this in our data (Appendix Fig. S3C) but can also add a temporal dimension: for complexes such as ribosomes, RNA polymerase, chaperonin (CCT) complex and others, the majority

of component subunits not only showed similar average heavy to total protein ratios but also a similar change in synthesis over the daily cycle (Fig. 3F; Appendix Fig. S3D,E). This supports the hypothesis that the assembly and turnover of macromolecular protein complexes is under circadian control.

Using an alternative approach to estimate the importance of rhythmicity for interactions of proteins within complexes, we took advantage of the STRING protein–protein interaction database (Szklarczyk et al, 2021). Unlike proteins with rhythmic synthesis at the whole-cell level, rhythmic proteins in this complex fraction had significantly more annotated physical interactions than would have been expected by chance given all proteins detected (Fig. 3G; Appendix Fig. S4). Importantly, these rhythmically synthesised protein subunits were almost all clustered within the same circadian phase (discussed below).

To validate these observations by an orthogonal method, we pulse-labelled cells with methionine analogue L-azidohomoalanine (Dieterich et al, 2006). AHA is an exogenous substrate, with a lower affinity for methionyl-tRNA synthetase than methionine, whose incorporation into polypeptide chains could potentially impact the stability of labelled proteins (Ma and Yates, 2018). We therefore only used AHA to assess nascent complex synthesis, rather than turnover. We analysed the incorporation of the newly synthesised, AHA-labelled proteins into highest molecular weight protein species detected under native-PAGE conditions (Fig. 3H; Appendix Fig. S3F). We observed a high amplitude daily rhythm of AHA labelling, indicating the rhythmic translation and assembly of nascent protein complexes. Taken together, these results show that daily rhythms in synthesis and degradation may be particularly pertinent for subunits of macromolecular protein complexes.

## Temporal consolidation of biological functions

Within the MMC fraction, we found that the vast majority of rhythmically synthesised proteins showed the highest synthesis at

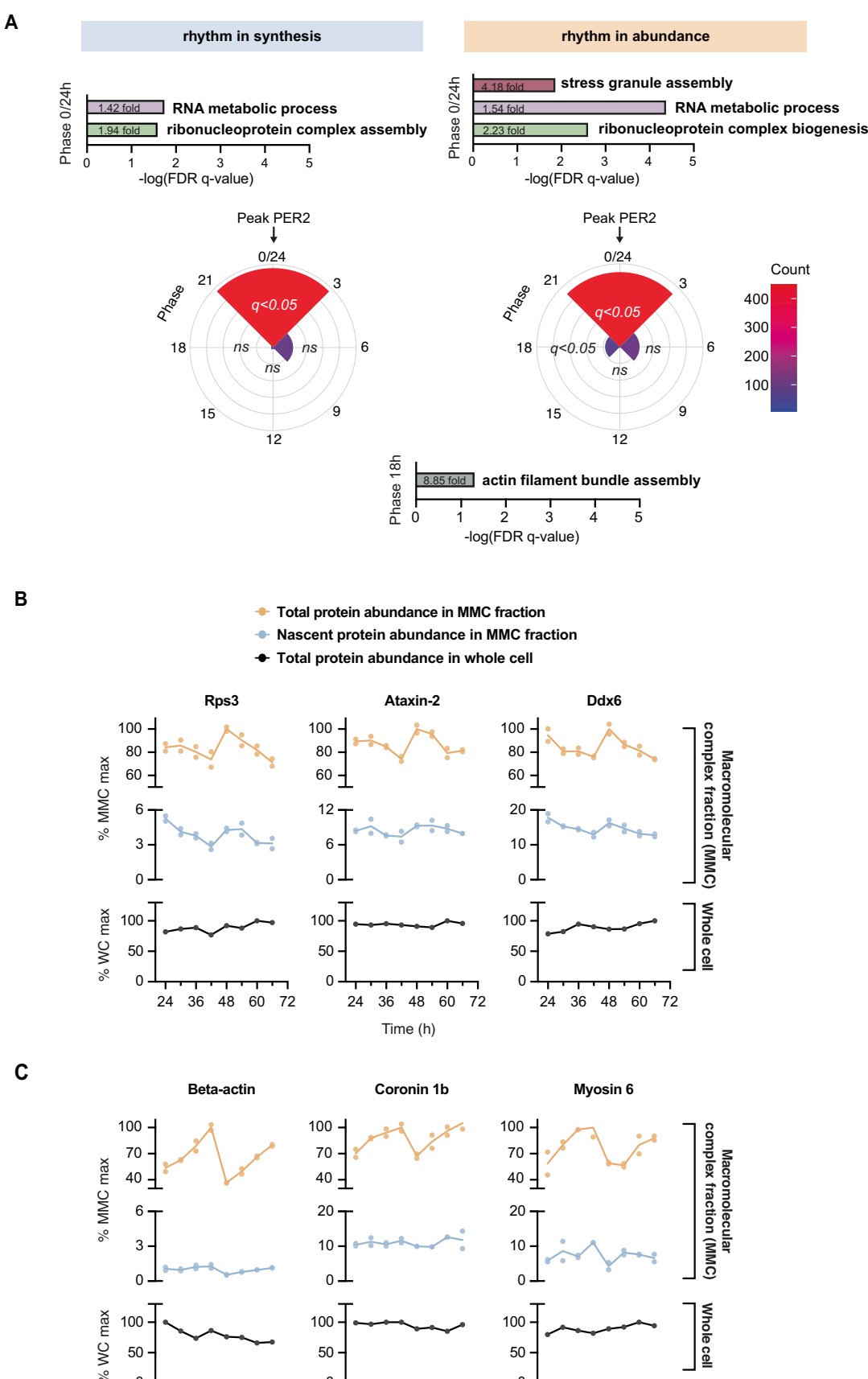

**Figure 4. Functional analysis of circadian complex assembly in cultured fibroblasts.**

(A) Phase distribution of proteins rhythmic (RAIN P < 0.05) in their total abundance and in their synthesis in the MMC fraction. Gene ontology functional enrichment for biological processes was performed using the GOrilla tool, with proteins at each phase compared against all detected proteins in this experiment, and FDR q-value (multiple comparisons adjusted P value) threshold of 0.05. For phase "0/24", top significant non-overlapping terms are presented, alongside their significance and fold-enrichment values. In phase "18", terms associated with actin were enriched (q < 0.05), e.g., "actin filament bundle assembly" (8.85-fold enrichment). (B) Example proteins peaking at phase "0/24", belonging to terms associated with ribonucleoprotein complex assembly and stress granule assembly. Total protein abundance (top) and nascent protein synthesis (middle) in the MMC fraction and total protein abundance at whole-cell level (bottom) is shown for each replicate, with line joining the mean at each timepoint. (C) As for (B), example proteins peaking at phase "18", associated with actin assembly. Source data are available online for this figure.

the same biological time, shortly after the peak of the PER2::LUC circadian bioluminescence reporter. Gene ontology analysis of these proteins (compared with a background list of all proteins detected in this fraction) revealed a clear enrichment for RNA binding proteins and terms associated with ribosome assembly (Fig. 4A). At the same circadian phase, rhythmically abundant proteins were similarly enriched for terms relating to RNA binding and ribonucleoprotein biogenesis, as well as many proteins associated with stress granule assembly, such as ataxin-2 and many DDX family members (Fig. 4A,B; Dataset EV2). Ribosomes and stress granules themselves control protein synthesis and regulate each other, so it is challenging to ascribe causal relationships between the two (Buchan and Parker, 2009; Riggs et al, 2020; Delarue et al, 2018) but our analysis clearly suggests a cell-autonomous surge of ribosome biogenesis.

Rhythmically synthesised/abundant proteins belonging to classes associated with ribonucleoproteins did not exhibit commensurate total abundance oscillations at the whole-cell level (Figs. 2 and 4B; Dataset EV1; (Wong et al, 2022; Hoyle et al, 2017)), and this might indicate that some abundance variation in the MMC fraction arises from redistribution between lighter and denser fractions over the circadian cycle, consistent with circadian regulation of protein solubility and compartmentalisation described previously (Wang et al, 2019; Stangherlin et al, 2021b; Jang et al, 2015; Malcolm et al, 2019; Zhuang et al, 2023). Supporting this possibility, we noted a smaller group of rhythmically abundant proteins in the phase preceding ribosome biogenesis, without any accompanying change in synthesis. These proteins were enriched by ninefold for actin and associated regulators of the actin cytoskeleton (q < 0.05, Fig. 4A,C). This is consistent with circadian regulation of cytoskeletal dynamics and actin polymerisation that we and others have described previously (Hoyle et al, 2017; Gerber et al, 2013). Indeed, as one of the most abundant cellular proteins, by mass alone, beta-actin accounted for 67% of the temporal compositional variation in the phase preceding ribosome biogenesis (Dataset EV2).

## Circadian regulation of ribosome turnover, not abundance

The ribosome is by far the most abundant macromolecular complex in the cell (An and Harper, 2020) and showed clear evidence of circadian regulation of turnover but not abundance at the whole-cell level in our pSILAC proteomics. To validate this result, we took advantage of two important observations: (1) all fully assembled ribosomes incorporate ribosomal RNA (rRNA) which can be readily separated from most other cellular RNA by density gradient centrifugation; (2) pulse-labelling with heavy uridine-$^{15}N_2$ allows nascent RNA to be distinguished from pre-existing RNA. RNA

could then be nuclease-digested, and the ratio of light to heavy uridine 5'-monophosphate (UMP) quantified by mass spectrometry. By combining this stable isotope labelling with ribosome purification, we developed a novel cellular assay which we could use to identify nascently assembled ribosomes (Fig. 5A). Circadian variation in the proportion of heavy UMP-containing assembled ribosomes, without an accompanying variation in total UMP abundance would directly demonstrate rhythmicity in ribosomal turnover.

In line with this prediction, over a circadian time series under constant conditions, we observed an ~24 h oscillation in the percentage of heavy UMP detected within assembled ribosomes (Fig. 5B). We found the highest rates of assembly occurred after the peak of PER2::LUC bioluminescence (Fig. 5B), in line with pSILAC (Fig. 3F). In addition, we observed an oscillation of heavy UMP in total RNA, likely due to rRNA which comprises >80% of total cellular RNA (Blobel and Potter, 1967; Palazzo and Lee, 2015). Importantly, total UMP within assembled ribosomes did not change significantly over time and nor did total cellular RNA (Fig. 5B), providing further evidence for circadian regulation of macromolecular complex turnover rather than abundance, in line with our MMC fraction pSILAC results.

## Rhythmic response to proteotoxic stress in cells and in mice

Disruption of proteostasis and sensitivity to proteotoxic stress are strongly linked with a wide range of diseases (Wolff et al, 2014; Harper and Bennett, 2016; Labbadia and Morimoto, 2015; Hipp et al, 2019). Evidently, global protein translation, degradation and complex assembly are crucial processes for cellular proteostasis in general, so cyclic variation in these processes would be expected to have (patho)physiological consequences. Elevated levels of misfolded, unfolded, or aggregation-prone proteins perturb proteostasis and provoke proteotoxic stress responses that disrupt cellular function, leading to cell death unless resolved (Santiago et al, 2020; Deshaies, 2014). Informed by our observations, we predicted that circadian rhythms of global protein turnover would have functional consequences for the maintenance of proteostasis. Specifically, we expected that cells would be differentially sensitive to perturbation of proteostasis induced by proteasomal inhibition using small molecules such as MG132 and BTZ, depending on time of day.

We first assessed the phosphorylation status of eIF2α, the primary mediator of the integrated stress response (ISR) pathway, throughout a full circadian timecourse in fibroblasts under unperturbed versus stress-induced conditions. As expected, acute proteasomal inhibition by 4-h treatment with MG132 induced eIF2α phosphorylation (Jiang and Wek, 2005), but importantly this induction varied depending on time of drug treatment (Fig. 6A;

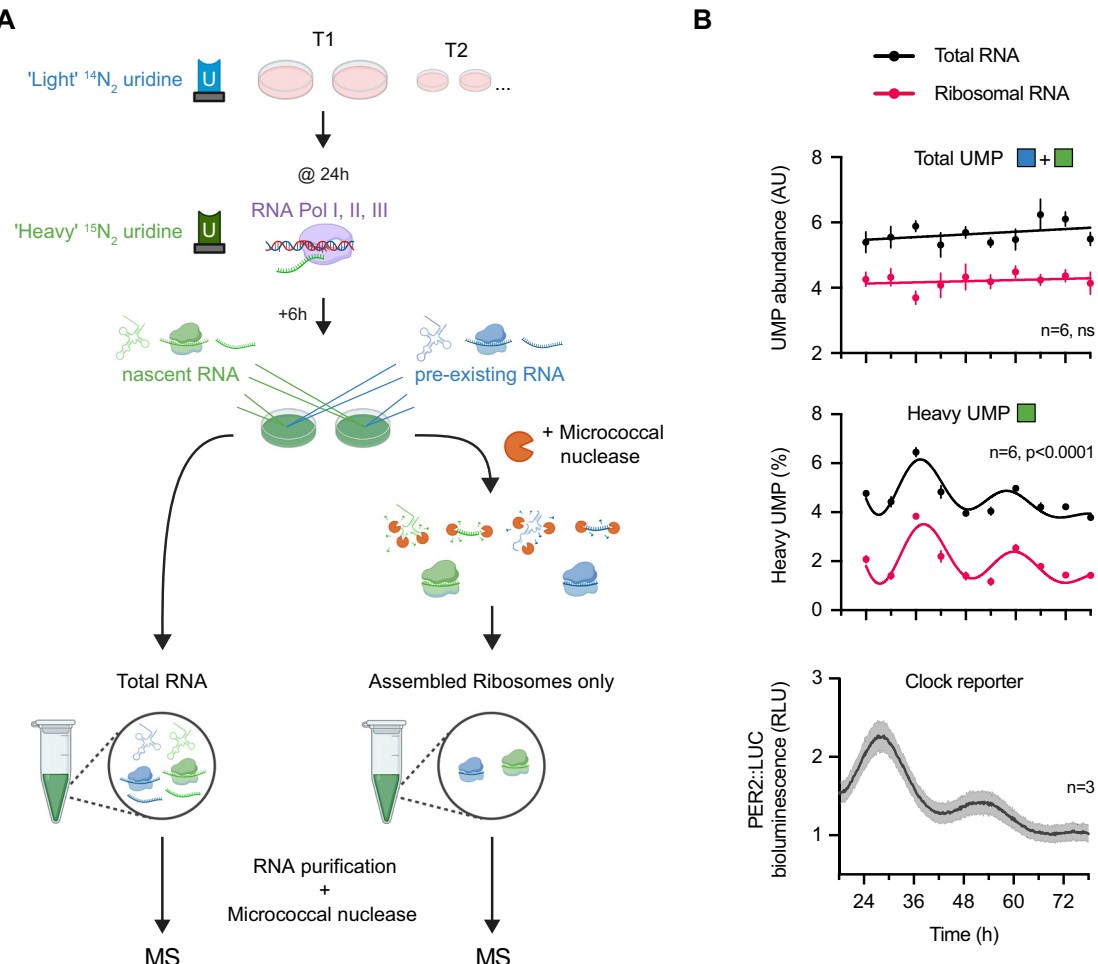

**Figure 5. Validation of circadian ribosome assembly in cultured fibroblasts by Edmondson assay.**

(A) Schematic of Edmondson assay for nascent rRNA labelling. In a set of entrained fibroblasts, at each circadian timepoint, 200 μM of heavy ($^{15}N_2$)-uridine was spiked into media for 6 h to metabolically label all nascent RNA prior to collecting cells. Total RNA or rRNA derived from assembled ribosomal complexes was extracted from cell pellets before RNA purification and treatment with micrococcal nuclease to completely digest RNA to free nucleotide monophosphates. The abundance of heavy, $^{15}N_2$, and light, $^{14}N_2$, uridine monophosphate (UMP) was subsequently quantified by mass spectrometry. (B) Circadian regulation of ribosome turnover detected by Edmondson assay. (Top) Total UMP abundance (heavy + light) for total RNA and rRNA from assembled ribosomes across time. (Middle) Proportion of heavy UMP for total RNA and rRNA from assembled ribosomes, expressed as a percentage of total cellular UMP and UMP that was incorporated within assembled ribosomes, respectively, across time. Heavy UMP reports RNA synthesised in the preceding 6 h. Note that heavy UMP was first corrected for its natural abundance detected in unlabelled fibroblasts. (Bottom) Parallel PER2::LUC bioluminescence recording, conducted under the same experimental conditions as the MS timecourse. Statistics: damped cosine wave fit compared with straight line (null hypothesis) by extra sum-of-squares F test, the statistically preferred fit is plotted and *P* value displayed. *n* = 6 (for UMP, top and middle) or 3 (for bioluminescence, bottom) replicate cell cultures. Data information: Data are presented as mean ± SEM of replicate cell cultures. Source data are available online for this figure.

Appendix Fig. S5A), with the highest fold-change increase observed around the predicted peak of protein turnover (shortly after the PER2::LUC peak). Phosphorylation of eIF2α leads to inhibition of canonical translation and is suggested to drive a daily decrease in bulk protein synthesis in vivo (Karki et al, 2020; Wang et al, 2019; Pathak et al, 2019). We did not observe any significant cell-autonomous rhythm in eIF2α phosphorylation under basal conditions (Appendix Fig. S5A), and so suggest that daily p-eIF2α rhythms observed in vivo likely arise through the interaction between cell-autonomous mechanisms and daily cycles of systemic cues, e.g., insulin/IGF-1 signalling and body temperature rhythms driven by daily feed/fast and rest/activity cycles respectively (Crosby et al, 2019; Beale et al, 2023a).

A major detrimental consequence of proteotoxic stress is the formation of insoluble intracellular protein aggregates (Albornoz et al, 2019; Dantuma and Lindsten, 2010). To test whether this was also time-of-day-dependent, we used a molecular rotor dye that becomes fluorescent upon intercalation into quaternary structures associated with protein aggregates (Shen et al, 2011) (Appendix Fig. S5B). As predicted, over 2 days, challenging cells with MG132 around the peak of protein turnover resulted in significantly more protein aggregation compared to controls than the same challenge delivered 12 h later (Fig. 6B; Appendix Fig. S5C).

Sustained proteotoxic stress results in cell death (Santiago et al, 2020; Deshaies, 2014), and cell death induced by 6 h treatment with BTZ showed a clear circadian rhythm (Fig. 6C). Strikingly, we

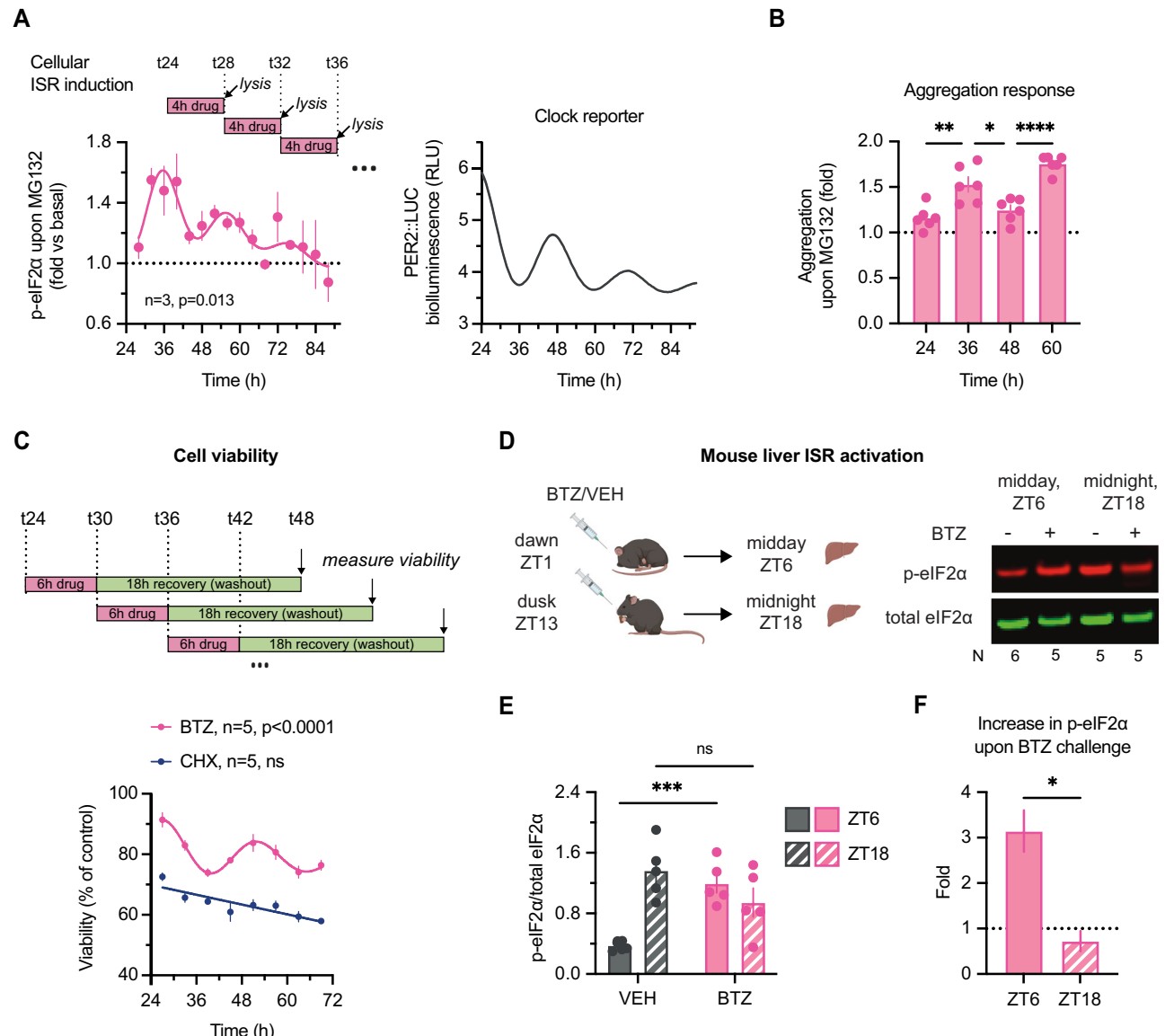

**Figure 6. Consequences of circadian regulation of protein homeostasis in culture and in vivo.**

(**A**) (Left) Two sets of fibroblast lysates were collected every 4 h for 3 days, one untreated control and one treated with 20 µM MG132 proteasomal inhibitor for 4 h before each collection. Fold-change increase in relative phosphorylation of eIF2α (i.e., (p-eIF2α/total)$_{MG132}$/(p-eIF2α/total)$_{control}$) at each timepoint is plotted. MG132 is expected to induce increase in phosphorylation of eIF2α but the extent of the induction differs. Statistics: damped cosine wave fit (plotted) preferred over straight line, extra sum-of-squares F test *P* value displayed. (Right) Parallel PER2::LUC bioluminescence recording, conducted under the same experimental and timecourse conditions as experiments in (**A–C**). (**B**) Cells were treated with 20 µM MG132 for 24 h starting at indicated timepoints, and aggregation relative to untreated samples measured by Proteastat aggresome kit. MG132 is expected to induce aggregation but the extent of the induction differs. Statistics: One-way ANOVA *P* < 0.0001; Dunnett's multiple comparison of neighbouring timepoints displayed on graph. The difference between T24 & T48 as well as between T36 & T60 was not statistically significant. *n* = 6 replicate cell cultures. (**C**) At eight timepoints throughout 2 days, fibroblasts were treated with 2.5 µM proteasomal inhibitor bortezomib (BTZ), 25 µM translation inhibitor cycloheximide (CHX), or vehicle control; after 6 h, the drugs were washed out, allowing cells to recover for further 18 h. Cellular viability after the treatments, as measured by PrestoBlue High Sensitivity assay, is expressed as a proportion of control (vehicle-treated) cells at each timepoint. Statistics: damped cosine wave fit compared with straight line (null hypothesis) by extra sum-of-squares F test, the statistically preferred fit is plotted & *P* value displayed. (**D**) Time-of-day bortezomib (BTZ) effect in vivo: mice received an i.p. injection of BTZ or vehicle control (VEH) at ZT1 or ZT13, and livers were harvested 5 h after the treatment. Representative Western blot is shown, blots, probed for total (green) and S51-phosphorylated (red) eIF2α. (**E**) Quantification of relative phosphorylation levels of eIF2α from experiment in (**D**). Statistics: two-way ANOVA, treatment × time interaction *P* = 0.0003; Sidak's multiple comparisons test between timepoints for each treatment is displayed. *N* = 24 mice, 5–6 mice/group. (**F**) Fold-change increase in phosphorylation of eIF2α upon bortezomib injection at each timepoint, quantified from (**E**) as the ratio of average p-eIF2α levels. Statistics: unpaired *t* test. *N* = 24 mice, 5–6 mice/group. Data information: In (**A–C**), data are presented as mean ± SEM of replicate cell cultures. In (**E, F**), data are presented as mean ± SEM of individual mice. **P* ≤ 0.05, ***P* < 0.01, *****P* < 0.0001. Source data are available online for this figure.

found roughly twice as much cell death occurred for proteasomal inhibition at the peak of protein turnover compared with its nadir (Fig. 6C; Appendix Fig. S5D). In contrast, translational inhibition with cycloheximide revealed no such temporal variation. Together, these data support our predictions, wherein proteasomal inhibition at peak times of translation and protein turnover exacerbates proteotoxic stress, protein aggregation, and cell death because the burden on protein quality control systems at these circadian phases is already high.

BTZ and its derivatives are used clinically to treat several types of blood cancers, associated with a multitude of side effects due to the proteasome's essential function in all cells (Deshaies, 2014; Manasanch and Orlowski, 2017; Zhang et al, 2020). In light of the daily variation in protein turnover we observed in mouse liver (Fig. 1F), we hypothesised that time-of-day sensitivity to BTZ would also be observed in vivo. Accordingly, we observed a stark day vs night difference in response to BTZ treatment in mouse liver, assessed by eIF2α phosphorylation (Fig. 6D–F; Appendix Fig. S5E). Consistent with this, time-of-day variation in BTZ-mediated inhibition of tumour growth was recently demonstrated in a mouse tumour model study (Wagner et al, 2021).

## Discussion

In this work, we provide evidence for coordinated circadian regulation of protein synthesis and degradation, resulting in rhythmic protein turnover, which is particularly significant for macromolecular complexes such as the ribosome. Across all experiments in this study, we find that protein synthesis, degradation and turnover is highest during the 6-8 h that follow maximal production of the clock protein PER2. This is coincident with increased glycolytic flux and respiration (Putker et al, 2018), increased macromolecular crowding in the cytoplasm, decreased intracellular $K^+$ concentration and increased mTORC activity (Feeney et al, 2016a; Stangherlin et al, 2021b; Wong et al, 2022). Just as temporal consolidation of protein synthesis is thought to increase its metabolic efficiency (O'Neill et al, 2020), we suggest that rhythmic turnover may serve to increase the efficiency of proteostasis by minimising deleterious changes in total cellular protein content and proteome composition.

The mechanistic underpinnings for cell-autonomous circadian regulation of the translation and degradation machineries remain to be fully explored, but are likely to be driven by daily rhythms in the activity of mTORC: a key regulator of protein synthesis and degradation as well as macromolecular crowding and sequestration (Stangherlin et al, 2021b, 2021a; Cao, 2018; Adegoke et al, 2019; Ben-Sahra and Manning, 2017; Delarue et al, 2018). In particular, global protein synthesis rates are greatest when mTORC1 activity is the highest, in tissues and cultured cells, whereas pharmacological treatments that inhibit mTORC1 activity reduce daily variation in crowding and protein synthesis rates (Feeney et al, 2016a; Lipton et al, 2015; Stangherlin et al, 2021b). Given our focus on proteomic flux and translation-associated protein quality control, autophagy was not directly within the scope of this study but is also mTORC-regulated and exhibits daily rhythms (Ma et al, 2011; Ryzhikov et al, 2019). In vivo, daily regulation of mTORC activity arises primarily through growth factor signalling associated with daily feed/fast cycles (Crosby et al, 2019; Byles et al, 2021). The mechanisms

facilitating cell-autonomous circadian mTORC activity rhythms are incompletely understood but may include Mg.ATP availability (Feeney et al, 2016a) and its direct regulation by PERIOD2 (Wu et al, 2019). This will be an important area for future work.

Increased translation will inevitably be associated with increased production of defective translation products, such as prematurely terminated or misfolded peptides that must be rapidly cleared by ubiquitin–proteasome system-mediated degradation (Dimitrova et al, 2009; Wang et al, 2013; Gandin and Topisirovic, 2014). Proteome-wide cycling ubiquitination sites have been recently described (Hansen et al, 2021); here we present evidence of cell-autonomous circadian rhythms of proteasome activity and rhythmic turnover for a greater proportion of the proteome than oscillates in abundance. Accordingly, temporal coordination was found for the synthesis of heteromeric protein complexes, in particular the ribosome, the most abundant protein complex in most mammalian cells. This highlights how, even though most mammalian proteins exhibit half-lives >24 h and show little daily variation in abundance, the rate at which they are replaced can be subject to circadian regulation. This may be particularly beneficial for heteromeric complex assembly. Within the MMC fraction, we observed enrichment for specific biological functions at different times of the day, e.g., ribonucleoprotein assembly vs actin polymerisation. While bulk measurements showed clear coordination on the global scale, data from whole-cell and fractionated proteomics suggest that a combination of rhythmic synthesis, degradation, crowding and sequestration acts in concert to temporally organise rhythmic macromolecular biogenesis and assembly whilst minimising changes in overall proteome composition.

More insight into the relationship between temporal organisation and proteostasis can be gained by comparing our findings with other model systems. For example, we recently found chronic proteostasis imbalance in cells and tissues deficient for the Cry1 and Cry2 genes, without which circadian regulation of transcription does not persist. These cells exhibit increased proteotoxic stress as well as increased circadian variation in proteome composition compared with wild-type controls (Wong et al, 2022). Moreover, the temporal compartmentalisation of proteome renewal processes has a clear precedent in yeast, where metabolic oscillations arise as a direct consequence of TORC-dependent cycles of protein synthesis and sequestration that are critical for preventing deleterious protein aggregation (O'Neill et al, 2020). In light of similar findings in the alga Ostreococcus tauri (Kay et al, 2021; Feeney et al, 2016a), we speculate that promoting and minimising the energetic cost of proteostasis may be an evolutionarily conserved function of circadian and related biological rhythms.

Beyond testing two key predictions in mouse liver, a limitation is that this study was restricted to quiescent primary mouse fibroblasts. In our experience, fibroblasts are a particularly powerful and predictive model for fundamental principles of cellular circadian regulation (Hoyle et al, 2017). Clearly though, in future it will be necessary to extend our initial findings of protein turnover in vivo to fully validate that daily rhythms of protein turnover and proteome renewal occur under natural conditions (daily light/dark, feed/fast, rest/activity cycles). We predict that they will be observed across multiple mature tissues, with higher amplitude than cultured cells due to amplification of cell-intrinsic processes by daily systemic cues (hormonal and body temperature rhythms). We anticipate that the relative phases of synthesis and degradation

rhythms will likely differ somewhat between tissues and physiological contexts, as recently found in growing muscle for example (Kelu et al, 2020).

Rhythms in transcription were not addressed in this study, but as discussed above, there is a well-established discrepancy between identities and phases of rhythmic proteins and their underlying transcript levels. Regulation at the translational level has been suggested to explain these differences, although ribosomal profiling studies have noted that on average there appears to be no delay between rhythmic transcript and nascent translation (Atger et al, 2015; Janich et al, 2015; Jang et al, 2015). We note, however, that ribosomal profiling reports on the level and position of ribosome-mRNA association, and so does not directly measure nascent protein production. Although a good correlate when comparing steady-state conditions, ribosome profiling also does not distinguish between active and stalled ribosomes, and does not reflect all the changes in protein synthesis that occur in dynamic cellular systems or upon perturbation that globally alter proteostasis (Liu et al, 2017). Upon finding evidence for global changes in protein synthesis and degradation throughout the day, the development of our pulsed SILAC method was crucial for allowing us direct insight into the regulation of protein abundance. Enabled by technological improvements in peptide detection accuracy and multiplexing, this is the first report of proteins tracked both across their lifetime (production) and across the circadian cycle. Moreover, our development of a simple pulse-labelling assay for nascent ribosome assembly likely has several applications beyond circadian research.

Finally, given the extensive links between proteome imbalance and many pathological states, daily regulation of protein metabolism has implications for health and disease. Circadian disruption is already strongly associated with impaired proteostasis, though causal mechanisms are poorly understood at this time (Bolitho et al, 2014; Musiek et al, 2018; Leng et al, 2019; Lipton et al, 2017; Wong et al, 2022). In this study, we predicted and validated that daily turnover rhythms confer daily variation on the sensitivity of cells and tissues to a clinically relevant proteasome inhibitor. This highlights how preclinical models may help to accelerate the development of (chrono)therapies, that optimise treatment outcomes by leveraging understanding of the body's innate daily rhythms (Cederroth et al, 2019).

# Methods

## Cell culture and general time course structure

Fibroblasts originated from mice homozygous for PER2::LUCI-FERASE (Yoo et al, 2004), isolated from lung tissue and were immortalised by serial passaging as described previously (Seluanov et al, 2010), and verified mycoplasma free by Lonza MycoAlert mycoplasma detection kit (LT07, Lonza). For routine culture, cells were maintained at 37 °C and 5% $CO_2$ in Gibco™ high glucose Dulbecco's Modified Eagle Medium (DMEM), supplemented with 100 units/ml penicillin and 100 µg/ml streptomycin, as well as 10% Hyclone™ III FetalClone™ bovine serum (GE Healthcare). When plated for experiments, cells were grown to confluence prior to the start of assaying, which ensures contact inhibition and elimination of cell division effects during the experiments (Hoyle et al, 2017; Ribatti, 2017).

For all the timecourse experiments, cells were subject to temperature entrainment, consisting of 12 h:12 h cycles of 32 °C:37 °C, for at least 4 days prior to the start of assaying, with media changes if required. Unless stated otherwise, the final medium change, containing 10% serum, occurred at the anticipated transition from 37 °C to 32 °C, as the cells were transferred to constant 37 °C. This is denoted as experimental time $t = 0$, or start of constant conditions. Sampling began at least 24 h afterwards (i.e., $t = 24 +$), to avoid any transient effects of the last serum-containing medium change and temperature shift (Balsalobre et al, 1998; Buhr et al, 2010; Beale et al, 2023b). Parallel recording of PER2::LUCIFERASE activity were obtained using ALLIGATOR (Cairn Research) (Crosby et al, 2017), and luminescence quantified in Fiji/ImageJ v2.0 (Abramoff, 2007; Schindelin et al, 2012).

## Cell lysis and protein quantification

For timecourse experiments requiring cell lysate collections, the procedure was based on the following. Cells were washed twice with PBS, and incubated with the indicated lysis buffers: normally either digitonin buffer (0.01% digitonin, 50 mM Tris pH 7.4, 5 mM EDTA, 150 mM NaCl) for 10 min on ice or urea/thiourea buffer for 20 min at room temperature (7 M urea, 2 M thiourea, 1% sodium deoxycholate, 20 mM Tris, 5 mM TCEP), both supplemented with protease and phosphatase inhibitor tablets (Roche, 4906845001 and 04693159001) added shortly beforehand. Cells were then scraped, and lysates transferred to Eppendorf tubes, before sonication with Bioruptor sonicator (Diagenode) at 4 °C, for 2–3 cycles 30 s on/30 s off. Lysates were then centrifuged at 14,000 rpm for 5 min, and the supernatant either flash-frozen in liquid nitrogen for future use, or taken directly for further analysis. For determination of protein concentration, Pierce bicinchoninic acid assay (BCA, Thermo Scientific, 23225) (Smith et al, 1985) was performed in microplate format according to manufacturer's instructions, with bovine serum albumin (BSA) protein standards diluted in the same lysis buffer as experimental samples. Pierce 660 nm assay (Thermo Scientific, 22660) was performed instead of BCA when samples contained thiourea.

## $^{35}$S pulse-chase labelling

All procedures for $^{35}$S pulse-chase were optimised to avoid methionine starvation, serum-containing media changes, and temperature perturbations, all of which could potentially reset circadian rhythms and obscure any cell-autonomous regulation. Fibroblasts were adapted to serum-free but otherwise complete medium starting from the last 4 days of temperature entrainment. At each timepoint, the cells were pulsed with 0.1 mCi/ml $^{35}$S-L-methionine and $^{35}$S-L-cysteine mix (EasyTag™ EXPRESS35S Protein Labeling Mix, Perkin Elmer) in methionine- and cysteine-free DMEM for 15 min. For chase, the radiolabel-containing media were replaced with standard DMEM supplemented with 2 mM (10× normal concentration) of non-radiolabelled methionine and cysteine, and cells incubated for 1 h. Throughout both pulse and chase the cells were maintained at 37 °C. At the end of pulse and chase periods, cells were washed with ice-cold PBS and lysed in digitonin buffer (0.01% digitonin (Invitrogen), 50 mM Tris pH 7.4, 5 mM EDTA, 150 mM NaCl for 10 min on ice). Lysates were run on NuPage™ Novex™ 4–12% Bis-Tris protein gels; the gels were then

stained with Coomassie SimplyBlue™ SafeStain (ThermoFisher). Gels were then dried at 80 °C for 45 min and exposed overnight to a storage phosphor screen (GE Healthcare, BAS-IP SR 2025), which was subsequently imaged with Typhoon FLA700 gel scanner and quantified in Fiji/ImageJ.

## Puromycin labelling

Puromycin dihydrochloride, diluted in PBS, alone or in combination with BTZ, was added directly to cells in culture medium, as 10× bolus to a final concentration of 1 µg/ml puromycin and 1 µM BTZ. Labelling proceeded for 30 min at 37 °C, after which cells were lysed in a urea/thiourea buffer and puromycin detected by western blotting.

## AHA incorporation

At each timepoint, while still maintaining cells at 37 °C, complete DMEM medium was replaced with methionine-free DMEM supplemented with AHA in combination with methionine at 30:1 ratio (Bagert et al, 2014)—1 mM AHA, 33 µM Met—and 1% dialysed FBS for 90 min. Cells were lysed in digitonin buffer (HEPES rather than Tris-buffered). AHA-containing proteins were conjugated to biotin by click chemistry, by adding appropriate reagents (Jena Bioscience) to the lysates, to final concentrations of 1 mM THPTA, 1 mM $CuSO_4$, 2 mM Na ascorbate, and 40 µM biotin alkyne, and incubating for 1 h at room temperature. Biotinylated proteins were then detected by western blotting.

## Western blotting

Samples for denaturing polyacrylamide gel electrophoresis (SDS-PAGE) were prepared by diluting lysates with reduced NuPage™ LDS sample buffer and heating at 70 °C for 10 min. Samples were run on NuPage™ Novex™ 4–12% Bis-Tris protein gels in MES buffer or on E-PAGE 8% 48-well gels (ThermoFisher). For native running conditions, NuPAGE Tris-Acetate 3–8% gels were used, with buffers as per the manufacturer's instructions.

For western blotting for puromycin and AHA incorporation measurement, chemiluminescence detection was used. Proteins were transferred from the gels to nitrocellulose membranes using an iBlot system (ThermoFisher). Membranes were stained by Ponceau as control for total protein loading, then washed, blocked, and incubated with primary antibody in the blocking buffer at 4 °C overnight. Anti-puromycin antibody (PMY-2A4-2 from Developmental Studies Hybridoma bank, at 1:1000) was used with 5% milk in TBST blocking buffer, and an anti-mouse HRP-conjugated secondary antibody, while AHA-biotin was detected using Strep-HRP antibody (R-1098-1 from EpiGentek at 1:2000) in 1% BSA, 0.2% Triton X-100 PBS blocking buffer, with additional 10% BSA blocking step before detection. Immobilon reagents (Millipore) were used to detect chemiluminescence. Images were analysed by densitometry in ImageLab v4.1 (BioRad).

For western blotting of total and p-eIF2α, LICOR protocols and reagents were used. Briefly, methanol-activated PVDF-FL (Immobilon) membranes were utilised for transfer, and dried for 1 h before blocking. After re-activation, membranes were blocked in Intercept TBS buffer. Primary (AHO0802 from ThermoFisher, ab32157 from Abcam, both at 1:1000) and secondary (IRDye 680RD and IRDye 800CW) antibodies were diluted in Intercept

TBS buffer with the addition of 0.2% Tween-20. Fluorescence was detected and quantified in Odyssey® CLx Imaging system.

## Proteasome activity assays

Cell-based ProteasomeGlo™ chymotrypsin-like and trypsin-like assays (Promega) were performed according to the manufacturer's instructions (Moravec et al, 2009) at multiple circadian timepoints as indicated. Briefly, cells in 96-well plates and the assay reagent were equilibrated to room temperature, before reagent addition, mixing, incubation for 10 min, and luminescence measurement with Tecan Spark 10 M plate reader, with integration time of 1 s per well. For analysis, bioluminescence from negative control wells (containing only culture medium and the assay reagent, but no cells) was subtracted from all the experimental conditions.

## Aggregation assays

PROTEOSTAT® Aggresome Detection kit (Enzo Life Sciences) was used for detection of protein aggregates (Shen et al, 2011). Cells in 96-well plates were treated with MG132 (as indicated in the figure legends), added as 10× bolus diluted in serum-free DMEM to the pre-existing culture media, and gently titurated, to avoid cellular rhythms resetting. Cells were permeabilised and stained simultaneously with PROTEOSTAT® dye and Hoechst 33342, as per the kit manufacturer's manual. Total fluorescence in blue and red channels, and representative images of individual wells were acquired using Tecan Spark Cyto plate reader.

## Viability assays

PrestoBlue™ High Sensitivity reagent (ThermoFisher), a resazurin-based dye, was used to measure cellular viability (Boncler et al, 2014; Xu et al, 2015). Cells in 96-well plates were treated with drugs or DMSO (vehicle) controls, as indicated in figure legends, added as 10× bolus diluted in serum-free DMEM on top of existing culture media. For drug washout in the timecourse experiments, cell medium was replaced with 1% serum DMEM, to allow recovery for 18 h. The assay was then performed in line with the manufacturer's guidelines: following PrestoBlue reagent addition and incubation at 37 °C for 20 min, fluorescence was measured in a Tecan Spark 10 M plate reader, with excitation at 550 nm and emission at 600 nm.

## Proteomics data collection and analysis

### Cell culture and sample collection for pSILAC-TMT

For pSILAC-TMT experiments, mouse lung fibroblasts were cultured in 10% dialysed FBS (dFBS). SILAC labelling was conducted in DMEM supplemented with 1% dialysed FBS and heavy-labelled amino acids instead of their light analogues, specifically 84 mg/L $^{13}C_6^{15}N_4$ L-Arginine and 146 mg/L $^{13}C_6^{15}N_2$ L-Lysine (Ong et al, 2002). In the first timecourse pSILAC experiment, sets of cells were labelled for 6 h, every 6 h over 2 days, and total cell lysates were extracted in urea/thiourea-based buffer ($n = 1$ per timepoint). In the second timecourse experiments, labelling was done for 1.5 h, every 6 h over 2 days in duplicates, and fractionation was performed as described below. For the booster channel, a fully heavy-labelled sample was used, where cells were cultured in DMEM with heavy amino acids for five passages

(3–4 w) but otherwise processed in the same way as the timecourse samples. Fractionation was based on LOPIT-DC protocol (Geladaki et al, 2019). Two 15-cm dishes per sample were used, cells were scraped in ice-cold PBS, centrifuged, and then lysed on ice by resuspension in a mild buffer (0.25 M sucrose, 10 mM HEPES pH 7.4, 2 mM EDTA, 2 mM magnesium acetate, protease inhibitors) and passaged through a Dounce homogeniser. Lysates were moved to thick-wall ultracentrifuge tubes (Beckman 343778 11 mm/ 34 mm) and centrifuged at $79,000 \times g$ for 43 min to pellet membranes and organelles. The supernatant was then centrifuged again at $120,000 \times g$ for 45 min. The resulting pellet was resuspended in 8 M urea 20 mM Tris buffer, and processed for mass spectrometry analysis ($n = 2$ per timepoint).

### Mass spectrometry analysis

**Protein digestion**: Protein samples were reduced with 5 mM DTT at 56 °C for 30 min and alkylated with 10 mM iodoacetamide in the dark at room temperature for 30 min. The samples were then diluted to 3 M urea and digested with Lys-C (Promega) for 4 h at 25 °C. Next, the samples were further diluted to 1.6 M urea and were digested with trypsin (Promega) overnight, at 30 °C. After digestion, an equal volume of ethyl acetate was added and acidified with formic acid (FA) to a final concentration of 0.5%, mixed by shaking for 3 min and centrifuged at $15,700 \times g$ for 2 min. The top organic layer was removed and the bottom aqueous phase was desalted using homemade C18 stage tips (3 M Empore) filled with porous R3 resin (Applied Biosystems). The stage tips were equilibrated with 80% acetonitrile (MeCN) and 0.5% FA, followed by 0.5% FA. Bound peptides were eluted with 30–80% MeCN and 0.5% FA and lyophilised.

**Tandem mass tag (TMT) labelling**: Dried peptide mixtures (50 μg) from each condition were resuspended in 24 μl of 200 mM HEPES, pH 8.5. 12 μl (300 μg) TMTpro 16plex or 18plex reagent (ThermoFisher) reconstituted according to the manufacturer's instructions was added and incubated at room temperature for 1 h. The labelling reaction was then terminated by incubation with 2.2 μl 5% hydroxylamine for 30 min. The labelled peptides were pooled into a single sample and desalted using the same stage tips method as above.

**Off-line high pH reverse-phase peptides fractionation**: In total, 200 μg of the labelled peptides were separated on an off-line, high-pressure liquid chromatography (HPLC). The experiment was carried out using XBridge BEH130 C18, 5 μm, $2.1 \times 150$ mm column (Waters), connected to an Ultimate 3000 analytical HPLC (Dionex). Peptides were separated with a gradient of 1–90% buffer A and B (A: 5% MeCN, 10 mM ammonium bicarbonate, pH 8; B: MeCN, 10 mM ammonium bicarbonate, pH 8, [9:1]) in 60 min at a flow rate of 250 μl/min. A total of 54 fractions were collected, which were then combined into 18 fractions and lyophilised. Dried peptides were resuspended in 1% MeCN and 0.5% FA, and desalted using C18 stage tips, ready for mass spectrometry analysis.

**Mass spectra acquisition**: The fractionated peptides were analysed by LC-MS/MS using a fully automated Ultimate 3000 RSLC nano System (ThermoFisher) fitted with a 100 μm × 2 cm PepMap100 C18 nano trap column and a 75 μm × 25 cm, nanoEase M/Z HSS C18 T3 column (Waters). Peptides were separated by a linear gradient of 120 min, 6–38% buffer B (80% MeCN, 0.1% FA). Eluted peptides were introduced directly via a nanoFlex ion source into an Orbitrap Eclipse mass spectrometer (ThermoFisher). Data

were acquired using FAIMS-Pro device, running MS3_RTS analysis, switching between two compensation voltages (CV) of −50 and −70 V.

MS1 spectra were acquired using the following settings: R = 120 K; mass range = 400–1400 $m/z$; AGC target = 4e5; MaxIT = 50 ms. The charge states 2–5 were included and dynamic exclusion was set at 60 s. MS2 analysis were carried out with collision induced dissociation (CID) activation, ion trap detection, AGC = 1e4, MaxIT = 35 ms, CE = 34%, and isolation window = 0.7 $m/z$. RTS-SPS-MS3 was set up to search Uniport *Mus musculus* proteome (2021), with fixed modifications cysteine carbamidomethylation and TMTpro at the peptide N-terminal. TMTpro K, Arg10 (R + 10.008), TMTpro K + K8 (K + 312.221) and methionine oxidation were set as dynamic modifications. Missed cleavages were allowed, and maximum variable modifications was set at 3. In MS3 scans, the selected precursors were fragmented by high-collision dissociation (HCD), and analysed using the orbitrap with the following settings: isolation window = 0.7 $m/z$, NCE = 55, orbitrap resolution = 50 K, scan range = 110–500 $m/z$, MaxIT = 200 ms, and AGC = 1e5.

**Raw MS data processing**: The acquired 18 raw files from LC-MS/MS were each split into two individual spectra, one with CV = −50 V and one with CV = −70 V, total 36 files, using FreeStyle software (ThermoFisher). These files were then processed using MaxQuant (Cox and Mann, 2008) with the integrated Andromeda search engine (v1.6.17.0). MS/MS spectra were quantified with reporter ion MS3 from TMTpro experiments and searched against UniProt *Mus musculus* Reviewed (Nov 2020) Fasta databases. Carbamidomethylation of cysteines was set as a fixed modification, while methionine oxidation, N-terminal acetylation, Arg10 and Lys8 were set as variable modifications.

### Data analysis

After the MaxQuant search, all subsequent proteomics data processing and analysis was performed in R (v3.6.1 and v4.1.2) with R Studio v1.2. The custom scripts are available via a GitHub repository, at https://github.com/estere-sei/circadian-pSILAC.

Peptide-level information from MaxQuant (evidence.txt output file) was used as a starting point. Contaminants and reverse hits were removed. Peptides were classified according to their labelling state: those that had at least one heavy arginine (Arg10) or lysine (Lys8) were classified as "heavy", and the rest were classified as "light". Entries for peptides with identical sequences in the same labelling state were grouped together (i.e., their reporter ion intensities across the 16 TMT channels were summed up), including peptides with other modifications such as methionine oxidation. Peptides with missing values were excluded. Total heavy label incorporation was quantified as overall proportion of summed intensities of heavy peptides over total summed intensities per TMT channel. Sample loading normalisation was performed, applying a scaling factor to equalise total summed intensity across TMT channels.

Peptides were filtered to leave only those that were detected in both heavy and light form. Peptide intensities belonging to the same leading razor protein accession were summed up to get total protein abundance value, while the sum of heavy peptides only for each protein represented the amount of synthesis. The ratio of heavy to total protein intensity averaged across the 8 timepoints was used to estimate relative turnover.

Several methods were used to assess the likelihood of significant circadian change over time in proteins' total abundance and synthesis, including Rhythmicity Analysis Incorporating Non-parametric Methods (RAIN) (Thaben and Westermark, 2014) and ANOVA. With RAIN, the data were tested for rhythms with period length of 24 h. For ANOVA, the data were log-transformed, and 2 days of sampling were treated as replicates. Oscillation phase was taken from RAIN outputs, and represented circadian time of the peak of oscillation, where time 0 is equivalent to the peak of PER2::LUC from parallel recordings. The extent of change over time is expressed as fold change, taking average ratio of peak to trough intensity values across the 2 days of sampling.

For protein complex membership analysis, a list was taken from Ori et al, 2016 which combined CORUM, COMPLEAT and manually annotated complexes and their subunits (Ori et al, 2016; Giurgiu et al, 2019; Vinayagam et al, 2013). Ensembl gene identifiers were converted from human to mouse by g:Profiler g:Orth tool (Raudvere et al, 2019), and matched with detected proteins. To assess variability of complex turnover, an analysis similar to one in Mathieson et al, 2018 was performed: standard deviation of the average relative turnover was calculated between proteins belonging to each detected complex, taking only complexes with more than four subunits, and compared to a dataset of the same size and structure but with proteins chosen randomly from all detected proteins (i.e., same number of complexes with same number of subunits as in annotated data but "subunits" chosen by random sampling).

For gene ontology functional enrichment analysis, GOrilla tool was used (Eden et al, 2009), comparing target protein list with all detected proteins as background, and setting FDR q-value cutoff at 0.05. REVIGO (0.4) was used to remove redundant terms. For analysis of protein–protein interactions, STRING web app was used (Szklarczyk et al, 2021), filtering for high-confidence physical interactions, and looking for enrichment against the background of detected proteins.

## Mouse tissue experiments

All animal work was licensed by the Home Office under the Animals (Scientific Procedures) Act 1986, with Local Ethical Review by the Medical Research Council and the University of Cambridge, UK. Throughout the experiments, wild-type C57 mice were housed in 12:12 h light:dark conditions.

For in vivo turnover measurements, mice received i.p. injections of either 40 µmol/kg puromycin (Ravi et al, 2020, 2018; Schmidt et al, 2009), or 40 µmol/kg puromycin in combination with 2.5 mg/kg BTZ (Apex Bio). Both solutions were sterile-filtered in PBS with 1% DMSO. Animals were culled 45 min after, in the same order as injected, and livers collected and flash-frozen in liquid nitrogen. The procedure was performed twice on the same day, 1 h after the transition from dark to light (ZT1), and 1 h after the transition from light to dark (ZT13). Four age-matched male mice were used per condition.

For in vivo response to proteotoxic stress measurements, mice received i.p. injections of 2.5 mg/kg BTZ (Apex Bio) or vehicle control (1% DMSO in PBS, sterile-filtered). Animals were culled 5 h after, in the same order as injected, and livers collected and flash-frozen in liquid nitrogen. Injections were performed twice on the same day, 1 h after the transition from dark to light (ZT1), and 1 h after the transition from light to dark (ZT13), and data are

presented at the timepoint of harvest (i.e., ZT6 or ZT18). 6 age-matched male mice were used per condition. The number of biological replicates was informed by a trial experiment performed under similar conditions with fewer animals.

Tissues were homogenised in urea/thiourea lysis buffer in Precellys 24 Tissue Homogeniser (Bertin Instruments), using CK14 ceramic beads, for $3 \times 15$ s at 5000 rpm with 30 s breaks. Lysates were then cleared by centrifugation at 14,000 rpm for 5 min, followed by protein sample preparation and Western blotting as previously described.

## Edmondson assay for nascent rRNA labelling

At each timepoint, whilst maintaining fibroblasts at constant 37 °C, 200 µM isotopically heavy uridine ($^{15}N_{2-,}$ Cambridge Isotope Laboratories) was spiked into media for 6 h to label nascently transcribed RNA. After labelling, cells were harvested by trypsinisation, and pellets immediately flash-frozen and stored at −70 °C.

For ribosome extraction, each cell pellet was resuspended in 200 µl of lysis buffer (40 mM HEPES·KOH (pH 7.5), 75 mM KOAc·HOAc, 5 mM Mg(OAc)$_2$·HOAc, 1 mM CaCl$_2$, 10 µM Zn(OAc)$_2$·HOAc, 2 mM spermidine, 5 mM dithiothreitol, 1% v/v Triton X-100) and sonicated at 4 °C for 5 min (30 s on; 30 s off). To each lysate, 2 µl of micrococcal nuclease (Nuclease S7, Roche) was added and the lysates were incubated at 25 °C for 18 min using a thermal cycler (Techne-Prime, Cole-Parmer). Processed lysates were immediately flash-frozen in liquid nitrogen and stored at −70 °C.

Ribosomal RNA samples and cell pellets for total RNA extraction were resuspended in 'RLT Buffer' and RNA extracted and purified using the RNeasy Mini kit (Qiagen, 74004) according to the manufacturer's instructions, including the on-column DNase-treatment (Qiagen, 79254). RNA was then further purified via an overnight ethanol precipitation at −20 °C, and RNA pellets resuspended in pre-heated 'Physiological Buffer' (50 mM HEPES·KOH (pH 7.5), 100 mM KOAc·HOAc, 20 mM Mg(OAc)$_2$·HOAc). RNA was then degraded into single nucleotides by overnight room temperature incubation with micrococcal nuclease (Nuclease S7, Roche) supplemented with 1 mM CaCl$_2$. The digestion reaction was then terminated by flash-freezing, and samples stored at −70 °C.

## LC-MS analysis of UMP and heavy UMP in RNA lysates

In total, 10 µl of RNA lysate was diluted in 40 µl of 10 mM ammonium acetate and transferred to a 96-well skirted PCR plate (Starlab International, Hamburg, Germany) and covered with a silicone sealing mat (Axymat, Salt Lake City, Utah, USA) prior to mixed mode LC-MS analysis using an ACE Excel C18-PFP (pentafluorophenyl) column ($150 \times 2.1$ mm, 2.0 µm, Hichrom, Reading, Berkshire, UK). Mobile phase A consisted of water with 0.1% formic acid with 10 mM ammonium formate and mobile phase B was acetonitrile with 0.1% formic acid. For gradient elution mobile phase B was held at 0% for 1.6 min. followed by a linear gradient to 30% B over 4.0 min, a further increase to 90% over 1 min. and a hold at 90% B for 1 min. with re-equilibration for 1.5 min giving a total run time of 6.5 min. The flow rate was 0.5 mL/min and the injection volume was 3 µL. The needle wash used was 1:1 water: acetonitrile.

For MS analysis using the Q Exactive Plus (ThermoFisher Scientific, Hemel Hempstead, Hertfordshire, UK) a full scan of

60–900 *m/z* was used at a resolution of 70,000 ppm in positive ion mode. The source parameters were as follows: an auxiliary gas temperature of 450 °C, a capillary temperature of 275 °C, an ion spray voltage of 3.5 kV and a sheath gas, auxiliary gas and sweep gas of 55, 15 and 3 arbitrary units, respectively.

Samples were analysed using Xcalibur (Thermofisher, Version 4.2) and processed and integrated using the Qual Browser and Quan Browser tools within Xcalibur to target specific analytes. Ion chromatogram areas corresponding to light, $^{14}N_2$, UMP ($m/z$ 325.0431), and heavy, $^{15}N_2$, UMP (m/z 327.0372) were extracted. Heavy UMP abundance is expressed as a proportion of total UMP (light + heavy) abundance as: heavy UMP % = 100 * heavy UMP abundance/(heavy UMP abundance + light UMP abundance). All identifications of compounds were carried out using reference of the accurate mass and verified using standards purchased from Sigma Aldrich.

## Statistical analysis

Statistical tests were performed using GraphPad Prism (version 8 and 9, Graphpad Software Inc, La Jolla, CA) and R (v3.6.1 and v4.1.2) with R Studio v1.2, and are indicated in figure legends. *P* values are either reported in figures directly, or annotated with asterisks: *$P \leq 0.05$; **$P \leq 0.01$, ***$P \leq 0.001$; ****$P \leq 0.0001$, *ns* not significant, $P > 0.05$. Number of replicates are reported as *n* or *N* (for technical and biological, respectively) in the figures; error bars represent standard error (SEM) unless stated otherwise. Outliers were excluded by ROUT with a Q = 1% in GraphPad Prism. In cases where a comparison of fits was performed, determining whether the data are better described by a straight line or a cosine wave with circadian period, the following equation was used for the latter:

$$y = (mx + c) + ae^{kx} \cos \frac{2\pi x - r}{p}$$

Where *m* is the baseline, *c* is the offset from 0 in *y* axis, *a* is the amplitude, *k* is the damping rate, *r* is the phase, and *p* is the period, which was fixed at either 24 h or 25 h depending on the parallel PER2::LUC recording period.

## Data availability

Raw proteomics data and custom analysis scripts can be found at https://github.com/estere-sei/circadian-pSILAC. The mass spectrometry proteomics data have been deposited to the ProteomeXchange Consortium via the PRIDE (Perez-Riverol et al, 2021) partner repository with the dataset identifier PXD049176.

The source data of this paper are collected in the following database record: biostudies:S-SCDT-10_1038-S44318-024-00121-5.

## Peer review information

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

## Acknowledgements

The authors thank all members of O'Neill lab, Rachel Edgar, Manu Hegde and Szymon Juszkiewicz for valuable feedback and discussions, as well as Kathryn Lilley and Holger Kramer for advice on proteomics. The authors also thank biomedical technical staff at Medical Research Council (MRC) Ares facility and LMB facilities for assistance. NMR was supported by the Medical Research Council (MR/S022023/1). JON was supported by the Medical Research Council (MC_UP_1201/4).

## Author contributions

**Estere Seinkmane**: Conceptualisation; Data curation; Software; Formal analysis; Investigation; Visualisation; Methodology; Writing—original draft; Writing—review and editing. **Anna Edmondson**: Investigation; Methodology. **Sew Y Peak-Chew**: Investigation. **Aiwei Zeng**: Investigation. **Nina M Rzechorzek**: Investigation; Methodology; Writing—review and editing. **Nathan R James**: Methodology. **James West**: Formal analysis; Investigation. **Jack Munns**: Investigation; Writing—original draft; Writing—review and editing. **David CS Wong**: Investigation. **Andrew D Beale**: Data curation; Formal analysis; Visualisation; Writing—review and editing. **John S O'Neill**: Conceptualisation; Data curation; Supervision; Funding acquisition; Methodology; Writing—original draft; Project administration; Writing—review and editing.

Source data underlying figure panels in this paper may have individual authorship assigned. Where available, figure panel/source data authorship is listed in the following database record: biostudies:S-SCDT-10_1038-S44318-024-00121-5.

## Disclosure and competing interests statement

The authors declare no competing interests.

