## [Peer Review File · The EMBO Journal]

Circadian regulation of macromolecular complex turnover and proteome renewal

Estere Seinkmane, Anna Edmondson, Sew-Yeu Peak-Chew, Aiwei Zeng, Nina Rzechorzek, Nathan James, James West, Jack Munns, David Wong, Andrew Beale, and John O'Neill

Corresponding authors: John O'Neill (oneillj@mrc-lmb.cam.ac.uk) , Andrew Beale (abeale@mrc-lmb.cam.ac.uk)

Review Timeline:	Transfer from Review Commons:	3rd Nov 23
	Editorial Decision:	8th Dec 23
	Revision Received:	4th Apr 24
	Accepted:	15th Apr 24

Editor: Kelly Anderson

Transaction Report:

This manuscript was transferred to The EMBO Journal following peer review at Review Commons.

Review #1

1. Evidence, reproducibility and clarity:

Evidence, reproducibility and clarity (Required)

The manuscript "Circadian regulation of protein turnover and proteome renewal" investigates the role of protein degradation in the circadian control of proteostasis. The researchers suggest that the relatively static levels of protein levels in a cell are incongruent with the known oscillation in protein synthesis. They therefore hypothesize that there should be a compensatory mechanism to counteract rhythmic protein synthesis, rhythmic protein degradation. To investigate this, they employ bulk pulse chase labeling to study the process of degradation. They identify a synchronization between the creation and turnover of proteins in a cell, implying the clock helps to maintain homeostasis through a novel mechanism. They note that these phases align with energy availability, granting a plausible reasoning behind the biological implementation of this regulation. In summary, this is a sound manuscript that adds to the research field. The experiments in this manuscript are well thought out, organized, and explained. In general, the authors do not go further in their conclusions than I think is warranted given the data that they have, though I think that there are some key items that should be addressed before the publication of this manuscript.

****Major notes:****

1. In figure 1, a clearer idea of what the ** means would be appreciated. What was the standard of significance for this measure?
2. In Figure 1b, it is important to note clearly in the text that this is not a direct measure of protein degradation, but a subtractive proxy. Though I don't think that necessarily makes the authors conclusions incorrect, the same result could also be obtained if an extra 15% of the proteins were moved into the insoluble fraction. This is the same for Figure 1E and F.
3. In figure 1c, is the noted oscillation in protease activity due to the oscillation of these proteins? What are the predicted mechanisms behind this? I don't think that this is necessarily within the scope of this paper but should be addressed in the discussion. Also, the peak degradation rate from Figure 1B is 4 hours before the peak enzyme activities. How can this observation be reconciled?
4. For the pSILAC analysis, the incorporation scheme has a six-hour window between the comparison of the light and heavy peptides. This makes it somewhat difficult to assess whether you are looking a clock effect from T1 or T1+6. This does not negate the findings, but it does question when the synthesis is occurring and what is being compared, which I think should be more clearly discussed in the manuscript. This is discussed later in the manuscript but should be mentioned in this section.
5. There are no error bars on figure 2C. What the pSILAC just done in a singlet? If so, the rhythms estimation is likely a large overestimate and should be noted.
6. Why were the genes selected in 2C? these are not discussed anywhere else in the manuscript.
7. The authors note that for Figure 2 "These observations are consistent with widespread rhythmic regulation of protein degradation." However, only 5-10% of the proteome is oscillating at any level and less with a discrepancy between synthesis and abundance, so "widespread" is an exaggeration and this statement should be limited to the degradation in the

rhythmic proteome.

8. The authors note that their more developed strategy in figure 3 would allow for the detection of less abundant proteins. However, they do not discuss that they in fact found less proteins overall, or if they were able to detect proteins of lower abundance. This is of some concern in determining if this is indeed the better method that they predict. How can the authors reconcile this issue? How can they rationalize this explains their increase in oscillating elements?

9. In the comparison of complex turnover rates, the authors need to provide a metric that backs their statement that "the majority of component subunits not only showed similar average heavy to total protein ratios but also a similar change in synthesis over the daily cycle" for figure 3F.

10. In reference to the AHA incorporation, why is the hypothesis not that, like the puramycin, you would not see oscillation unless you add BTZ? Shouldn't the active degradation regulate the incorporation of AHA such that there is no visible rhythm unless you suppress degradation?

11. The authors claim that there is enrichment of the actin cytoskeleton, but where this data can be found should be explained. The only thing that is shown is a few selected graphs of proteins in this pathway.

12. The authors note an oscillation in the total levels of p-eif2, commenting that these do not arise from the rhythms in total eif2a but temperature and feeding rhythms. However, unless I misunderstood, this work was done in fibroblast cell culture, so in this case, where would these temperature and feeding rhythms come from?

13. In Figure 5d, the treatment impeding degradation is causing cell death while the inhibition of translation does not. However, wouldn't too much, or not enough, translation, without compensatory regulation from degradation cause a problem in the same way that degradation does?

2. Significance:

Significance (Required)

The information that stems from this work is relevant and of interest to circadian clock field as how the regulation of the output of the circadian clock is implemented is still a major question in the field. This manuscript suggests a novel and plausible method for how, at least in part, this regulation occurs. However, the manuscript uses methods that do not measure degradation directly, which is a minor limitation. In addition, the mechanisms by which this regulation is imparted are not addressed in any meaningful way, even in the discussion.

3. How much time do you estimate the authors will need to complete the suggested revisions:

Estimated time to Complete Revisions (Required)

(Decision Recommendation)

Between 3 and 6 months

No

Review #2

1. Evidence, reproducibility and clarity:

Evidence, reproducibility and clarity (Required)

****Summary:****

This is a very interesting and well written paper that addresses key questions in the circadian organization of proteostasis. The paper investigates origins of cellular circadian rhythms, invoking a premise early that there is a poor correlation between rhythmic gene expression - regulated by the canonical TTFL - and rhythms of the proteome, which are rather meager. Specifically, they ask how a relatively stable proteome is possible if cells engage in rhythms of cellular protein synthesis? Their hypothesis is that protein degradation must rhythmically compensate for rhythms of synthesis and much of the manuscript is focused on defining the relationship between rhythmic global synthesis and rhythmic degradation. They employ a series of detailed proteomic investigations and biochemical assessments of protein synthesis coupled with various circadian reporters to assess proteasome function. The proteomic experiments reveal a limited number of proteins with oscillations in either synthesis or abundance or both and no discernible pathway organization however, a followup and more refined study that utilized fractionated samples and boosted heavy SILAC identified strikingly, that many proteins in relatively heavy fractions are rhythmic and that these fall into possible complexes including ribosome and chaperonins. Finally, they perform in vivo experiments testing whether the timing of proteotoxic stimuli regulates the degree of the integrated stress response measured as pEif2a.

Overall, I think that this is a fascinating paper that addresses an important question but falls short on mechanistically unifying them and completely contextualizing the findings in light of the canonical modes of circadian timekeeping leaving us with an important, but mostly descriptive set of findings. In addition, there are a number of important questions about data interpretation, some issues with data quality that should be addressed outlined below. With revision and further explication, this study will be an excellent addition to the growing field of circadian organization of the cellular proteome.

****Major and minor Comments.****

Figure 1.

Fig 1a. The difference in Pulse and Chase at ZT24 does not appear to reflect the quantified data in 1b. This should be reconciled to make the figure convincing.

How was the timing of the chase collection determined?

Fig 1d-e. What is the evidence that puromycin labeling results in 'rapid' turnover. Fig 1e seems to be missing the data from the treated and untreated conditions? How are the lines produced (e.g. linear versus rhythmic? Are these drawn lines or actual regressions?).

Why was 30 minutes chosen as labeling time? It seems hard to understand here how protein degradation kinetics can be measured by puromycin labeling if the authors' claim that puromycin labeling potentially changes degradation rates as a function - primary or secondary - of the labeling itself. It seems they are measuring the potential to degrade proteins.

How do they determine that they are measuring degradation of functionally relevant proteins as opposed to a host of premature truncations?

Fig 1e bottom - again is this a true regression line?

Perhaps two time points should be examined here - similar to the pulse chase performed with 35S labeling?

Fig 1f. It appears that Puro labeling results in a rhythm between ZT1 and ZT13 but no statistic is provided and appears that the 'ns' is the results of variance in the data as opposed to difference in means? - would this not contradict the cellular result? What accounts for the rhythm reversal in the presence/absence of BTZ.

While the authors have previously demonstrated an increase in rhythmicity of the proteome in Cry1/Cry2 double knockout cells, it would have been welcome here to test a global loss of circadian transcription in the degradation assay. One might expect that these rhythms would also be even higher. What I am really asking is: what is the mechanism for rhythmic degradation and is it dependent on the canonical clock?

Fig 2. How was the 'fixed window' timeframe determined?

Fig 3h. While admittedly difficult, the native PAGE is not of great quality and kind of unconvincing. Also not really sure why the AHA labeling is used here and nowhere else in the paper.

Fig 4. I was a little disappointed here that the authors did not directly assess macromolecular assembly of at least one of their "hits" and demonstrate functional relevance and most of the analysis is maintained at a very superficial, systemic level. STRING assemblies are not terribly helpful without clear k-means clustering or some other clearly visualizable metric for stratifying and organizing the putative PPI data - this figure (S3) could be markedly improved.

Is it possible that some macromolecular complexes have rhythms because their constituent proteins have differential half-lives when in one complex compared with another in circadian time? This possibility was not discussed.

Fig. 5.

Why is the first histogram in 3c not at unity?

Do ZT24 and Zt48 differ, similarly do ZT36 and ZT60?

Fig S4f is not of good quality with missing eIF2a total and therefore no loading controls. S4e? true regression lines?

While I thought these experiments were effective, they did not tie back well to the rest of the paper. What are the consequences of a temporally sensitive ISR? Which pathways does it effect in circadian time? Here, the main holes in this study are somewhat exposed; namely, a lack of mechanistic depth in explaining the very fascinating, albeit mostly descriptive, findings. The implicit assumption made here is that aggregation is 'bad' but could the opposite be just as true? Taking these considerations in account would further strengthen the discussion.

2. Significance:

Significance (Required)

This is a fascinating paper that addresses key questions in the circadian organization of the proteome. The paper's main findings are that rhythms of protein synthesis and degradation are temporally coordinated to maintain overall stability of the proteome in mouse fibroblasts. Furthermore, the authors present evidence that this temporal organization may be important for assembly of macromolecular complexes. While very interesting, the main limitations are a lack of biochemical and mechanistic explanation and evidence that verifies these, mostly descriptive, findings.

There are some relatively minor statistical and data quality issues that are probably addressable relatively quickly.

Upon revision the study would be a welcome addition to investigators interested in proteostasis, circadian biology, cell biology and proteomics.

I am a physician-scientist with expertise in circadian rhythms, cell biology, protein synthesis, and biochemistry.

3. How much time do you estimate the authors will need to complete the suggested revisions:

Estimated time to Complete Revisions (Required)

(Decision Recommendation)

Between 3 and 6 months

4. *Review Commons* values the work of reviewers and encourages them to get credit for their work. Select 'Yes' below to register your reviewing activity at Web of Science

Reviewer Recognition Service (formerly Publons); note that the content of your review will not be visible on Web of Science.

No

Review #3

1. Evidence, reproducibility and clarity:

Evidence, reproducibility and clarity (Required)

Seinkmane et al investigate circadian regulation of protein synthesis and degradation in cultured cells and in mice. Their main new finding is that protein synthesis and degradation are in many cases rhythmic but coordinated such that the proteome is rhythmically renewed without an apparent rhythm in total protein abundance. Particularly the pool of large protein complexes is rhythmically renewed in this fashion.

Using pulsed SILAC in combination with mass spectrometry, the authors are able to distinguish between total and newly synthesized protein levels in mouse lung fibroblasts. Analysis of these data shows that the synthesis of a large number of proteins is rhythmic although the total amount is constant, or that proteins are synthesized at a constant rate but the total amount is rhythmic, suggesting that degradation is rhythmic.

By analyzing macromolecular complexes, defined as a high-speed pellet, they also present evidence that the rhythmic components of large complexes oscillate in the same phase and have a similar protein turnover rate. The authors conclude that complexes assemble rhythmically.

The authors also present evidence that the activity of the proteasome oscillates in a circadian manner. Based on this observation, they show (in fibroblasts and in mice) that the response to proteotoxic stress (monitored by eIF2 α phosphorylation levels, protein aggregation, and apoptosis) is higher at circadian times of high proteasome activity.

I am an expert in the circadian field, and the hypothesis and concept behind the work presented here are potentially very interesting, and the experimental design is in principle suitable to answer these questions. However, after reading the paper several times, I cannot find the set of experiments that would convincingly support the authors' conclusions.

****Major questions/points:****

The major limitation of the manuscript is that the conclusions rely heavily on statistical analysis and massive processing of data from a bewilderingly large number of very different experiments. The critical experiments lack replicates and obvious controls. In looking at the figures, I have often wondered if the presence or absence of a rhythm is real or a product of the heavily processed data. The fact that a cosine wave fits through data points better than a

straight line does not necessarily mean that a circadian rhythm is present.

I think that in particular, the SILAC experiment(s) should be repeated and also performed with an arrhythmic control (such as CRY1/2 KO). Comparability between the whole cell and MMC SILAC experiments is also limited due to the different experimental conditions (6h vs. 1.5h pulse, +booster).

The essential and new message of the paper is that (at least some) macromolecular complexes undergo circadian renewal (degradation and synthesis). Rather than just analysing an operationally defined pellet fraction by mass spectrometry, this could be shown in more detail and directly for one or two specific macromolecular complexes. Ribosomes, for example, seem particularly suitable, because there would also be the very simple approach of measuring the synthesis of ribosomal RNA by pulse labelling. To me, such an analysis would be perfectly sufficient as a proof of principle. I would then omit aspects such as rhythmic stress response, since many additional experiments are needed to demonstrate this convincingly.

****Specific points:****

The reader is strongly influenced by the cosine wave or straight lines in the graphs (e.g. 1c, e, 3h, 5b, etc) produced by the analysis of rhythmicity, which basically only gives a yes or no answer. But it is not really that simple. If the algorithm detects a rhythm what is its period? Is it the same as the period of the luciferase reporter? If the period lengths correlate, do the phases as well (e.g. see differences in phases 1c and e)? These questions are not addressed.

The algorithm in Fig 1c predicts a rhythm for the chymotrypsin-like and the trypsin-like but not for the caspase-like activity. The peptide assay measures core proteasome activity independent of ubiquitylation and should therefore be dependent on proteasome concentration in the sample. How can then only two of the three proteasomal activities be rhythmic? Please elaborate and repeat with arrhythmic cells (e.g. CRY1/2 KO). The period length does not seem to correlate with the one of the reporter. Why is that?

Fig. 1a,b suggest that there is a rhythm in global protein synthesis with a significant peak at 40h. Yet, Fig. 1e suggests otherwise. How can that be? Also, the degradation graph (lower panel 1c) has to be plotted with the ratios calculated from the data points and not the heavily processed fitted graphs. This can be very misleading.

It also strikes me as odd that the amplitude of degradation increases (peak at 28h lower than at 30h) while the amplitude of the core clock oscillation dampens over time (peak at 54h higher than at 53h due to desynchronisation). Only two data values around 54h are responsible for the detected rhythm (2nd peak). Furthermore, phase and period do not agree with the rhythm of proteolytic activities shown in 1c. How can this be explained?

Regarding the MS data shown in Figure 2, is it possible to show a positive / quality control? Best would be MS data of Luciferase (or PER2,3, RevErb/alpha, DBP) to show oscillation of protein levels with the same phase and period as the reporter.

In Fig. 2c examples of the 4 groups of proteins presented in 2e should be shown (both synthesis and total abundance arrhythmic, either one rhythmic or both rhythmic) and not just what appears to be random examples of rhythmic and arrhythmic proteins.

Is it possible at all to distinguish between synthesis/turnover and assembly/disassembly of macromolecular complexes in the MMC SILAC experiment? If so, how?

Looking at Fig. 4b,c, what is the fraction of rhythmic proteins from the MMC experiment

that also oscillate in either synthesis, total abundance or both in the whole cell? Is there a general correlation at all? Please show.

Why is the phase of the oscillating proteins different in the two experiments (compare Figs. 2f,g and 4a) and does either of them match with the phase of the PER2::LUC reporter, which should be the peak synthesis phase of the clock?

Regarding the sensitivity to MG132 in Fig. 5b it doesn't make sense that, while eIF2alpha phosphorylation is arrhythmic in untreated cells and the levels of eIF2alpha phosphorylation are (apparently) not exhibiting a rhythmic change by administration of MG132 at different circadian timepoints, the ratio of P-eIF2alpha with and without MG132 suddenly is. Please show in Fig. S4b quantifications of the individual experiments with and without MG132. What is presented in 5b is after all the ratio of ratios of quantifications of Western blots, each of which individually does not display any appreciable rhythm. For me this is too much of processing of data. In my opinion, the MG132 4h acute treatment must show a detectable rhythm.

****Minor:****

In Fig. 1f please show dot blot with error bars as well as the individual experiments in the supplementals. Please check the graph legend (N>=3?)

Please explain the mechanism of the "booster" used in the second SILAC experiment.

p10 3rd paragraph: S2e not S3e

p12 last paragraph please add reference to Figs. 5f,g

2. Significance:

Significance (Required)

xxxxxx

3. How much time do you estimate the authors will need to complete the suggested revisions:

Estimated time to Complete Revisions (Required)

(Decision Recommendation)

Between 3 and 6 months

4. Review Commons values the work of reviewers and encourages them to get credit for their work. Select 'Yes' below to register your reviewing activity at Web of Science

Reviewer Recognition Service (formerly Publons); note that the content of your review will not be visible on Web of Science.

Yes

Full Revision

Manuscript number: RC-2022-01700

Corresponding author(s): John O'Neill, Andrew Beale

1. General Statements [optional]

We would like to thank the reviewers for their helpful comments and suggestions which have helped to improve the manuscript substantially. Indeed, we have updated the title of the manuscript to reflect a greater focus on the primary finding, that circadian rhythms function to minimise the bioenergetic cost of protein homeostasis, through temporal consolidation of protein turnover and in particular macromolecular complexes. This refinement of focus was stimulated thanks to reviewers' suggestion to validate the macromolecular proteomics with an orthogonal method for which we examined ribosomal turnover. As the most abundant macromolecular complex in the cell, accounting for >80% of total cellular RNA and 30-40% of total cellular protein, turnover of ribosome is of primary importance for cell-wide protein homeostasis. We have addressed all of their concerns as detailed in the point-by-point rebuttal below. We trust that you will find this revised version suitable for publication.

Please find below a point-by-point reply to the reviewers, with our comments marked in red. Text revisions are highlighted in yellow in the manuscript file; direct quotations of these revisions in the below point-by-point reply are in *"quotation marks and italics"*.

Reviewer #1 (Evidence, reproducibility and clarity (Required)):

The manuscript "Circadian regulation of protein turnover and proteome renewal" investigates the role of protein degradation in the circadian control of proteostasis. The researchers suggest that the relatively static levels of protein levels in a cell are incongruent with the known oscillation in protein synthesis. They therefore hypothesize that there should be a compensatory mechanism to counteract rhythmic protein synthesis, rhythmic protein degradation. To investigate this, they employ bulk pulse chase labeling to study the process of degradation. They identify a synchronization between the creation and turnover of proteins in a cell, implying the clock helps to maintain homeostasis through a novel mechanism. They note that these phases align with energy availability, granting a plausible reasoning behind the biological implementation of this regulation. In summary, this is a sound manuscript that adds to the research field. The experiments in this manuscript are well thought out, organized, and explained. In general, the authors do not go further in their conclusions than I think is warranted given the data that they have, though I think that there are some key items that should be addressed before the publication of this manuscript.

Thank you for reading and appreciating our work

Major notes:

1) In figure 1, a clearer idea of what the ** means would be appreciated. What was the standard of significance for this measure?

Thank you, this was already reported in the methods section but is now reported in the figure legend also.

2) In Figure 1b, it is important to note clearly in the text that this is not a direct measure of protein degradation, but a subtractive proxy. Though I don't think that necessarily makes the authors conclusions incorrect, the same result could also be obtained if an extra 15% of the proteins were moved into the insoluble fraction. This is the same for Figure 1E and F.

Considering only the pulse shown in the left-hand graph of 1B, the reviewer is correct that this could arise by rhythmic partitioning of nascently synthesised proteins between digitonin-soluble and insoluble fractions. This could not readily explain the variation in the % of nascently synthesised digitonin-soluble protein that is degraded however (right hand graph), hence the need for pulse-chase rather than pulse alone. As such, we do not exclude circadian-regulated solubility of nascently synthesised protein or that there is a rhythm of protein synthesis in the soluble fraction, both are likely true. Rather Figure 1B indicates the relative proportion of nascently-synthesised protein in the soluble fraction that is degraded within 1h of synthesis is not constant over time. This is consistent with current understanding of the regulated increase in activity of protein quality control mechanisms (including proteasome-mediated degradation) that are required to maintain protein homeostasis upon an increase in bulk translation (Gandin and Topisirovic, Translation, 2014).

In contrast, the lysates probed in Fig 1F were extracted in denaturing urea/thiourea buffer and so cannot be explained by variation in protein solubility.

Considering 1E, to explain this result entirely through solubility changes would require that puromycinylated polypeptides to become more soluble, at discrete phases of the circadian cycle, but only when the proteasome is inhibited. Whilst we cannot formerly exclude this possibility, we are not aware of evidence to support it, whereas there is prior evidence supporting circadian regulation of protein synthesis and proteasome activity.

To communicate all of this more clearly we have made the following revisions to the text:

Page 6: *"The experiment was performed over a 24h time series followed by soluble protein extraction using digitonin, which preferentially permeabilises the plasma membrane over organelle membrane."*

Page 6: *"Importantly, the proportion of degraded protein varied over time, being highest at around the same time as increased protein synthesis (Fig 1B), indicating time-of-day variation in digitonin-soluble protein turnover which cannot be solely attributed to previously reported circadian regulation of protein solubility (Stangherlin et al, 2021b). Rather, it suggests that global rates of protein degradation may be co-ordinated with protein synthesis rates, and may vary over the circadian cycle."*

Fig 1a legend: "...with digitonin buffer"

Fig 1e legend: "...in digitonin buffer"

Fig1f legend: "... and extracted with urea/thiourea buffer"

3) In figure 1c, is the noted oscillation in protease activity due to the oscillation of these proteins? What are the predicted mechanisms behind this? I don't think that this is necessarily within the scope of this paper but should be addressed in the discussion. Also, the peak degradation rate from Figure 1B is 4 hours before the peak enzyme activities. How can this observation be reconciled?

Besides this study, our two previous proteomic investigations of the fibroblast circadian proteome detected no biologically significant or consistent rhythm in proteasome subunit abundance (Wong et al., EMBO J, 2021; Hoyle et al., Science Translational Medicine, 2017). Moreover, proteasomes are long-lived stable complexes whose activity is determined by a combination of substrate-level, allosteric and post-translational regulatory mechanisms that includes their reversible sequestration into storage granules (Albert et al., PNAS, 2020; Fu et al., PNAS, 2021; Yasuda et al., Nature, 2020). It is therefore very likely that the observed rhythm in trypsin- and chymotrypsin-like activity occurs post-translationally. Proteasome subunit composition is also known to change, which might be another reason for differences between the protease activities (Marshall and Vierstra, Front Mol Biosci, 2019; Zheng et al., J Neurochem, 2012).

Due to the nature of the experiment, the degradation rate inferred from Figure 1B does not reflect proteasome activity, exclusively. Rather it reflects the combined sum of processes that remove nascently produced proteins from the cell's digitonin-soluble fraction, which includes proteasomal degradation, but also autophagy, protein secretion and sequestration into other compartments. Therefore, the peak degradation in Fig 1B would not necessarily be expected to coincide with the peak of proteasome activity in Fig 1C. Figure 1A/B is intended as an exemplar for the investigation's rationale and was the first to be performed chronologically.

To communicate this succinctly, we have revised the relevant text as follows:

Page 7: "*Previous proteomics studies under similar conditions have revealed minimal circadian variation in proteasome subunit abundance (Wong et al, 2022), suggesting that proteasome activity rhythmicity, and therefore rhythms in UPS-mediated protein degradation, are regulated post-translationally (Marshall & Vierstra, 2019; Hansen et al, 2021)*"

4) For the pSILAC analysis, the incorporation scheme has a six-hour window between the comparison of the light and heavy peptides. This makes it somewhat difficult to assess whether you are looking a clock effect from T1 or T1+6. This does not negate the findings, but it does question when the synthesis is occurring and what is being compared, which I think should be more clearly discussed in the manuscript. This is discussed later in the manuscript but should be mentioned in this section.

Thank you for this suggestion. To communicate this more clearly, we have rearranged the labels at the top of schematic graphs in figures 2b and 3b in order to clearly distinguish the pulse-labelling window from the time of sample collection. The following text has been added to the methods section:

Page 9: *"To enable sufficient heavy labelling for detection, a 6h time window was employed, thus measuring synthesis and abundance within each quarter of the circadian cycle "*

5) There are no error bars on figure 2C. What the pSILAC just done in a singlet? If so, the rhythms estimation is likely a large overestimate and should be noted.

This first pSILAC experiment was performed in singlet with respect to external time for the RAIN analysis, but is duplicate for the two-way ANOVA that is also reported, by treating each cycle as a separate replicate. In fact, the 6.2% of proteins that were significantly rhythmically abundant by RAIN actually agree well with two previous experiments we performed using mouse fibroblasts under identical conditions: the first with 3h resolution over 3 cycles in singlet (7% rhythmic), the second with 4 biological independent replicates over one cycle (8% rhythmic) (Wong et al., EMBO J, 2021). The curve fits shown in 2C are the standard damped sine wave fits, with p-values from RAIN reported in the figure legend.

Most importantly however, and as noted in the text, the absolute % of rhythmically abundant proteins is rather irrelevant and indeed the absolute numbers of 'rhythmic' proteins can vary wildly, dependent on the analysis method and stringency. The only important point to be gleaned from the estimates shown in Figure 2e is that by either statistical test, most rhythmically abundant proteins are not rhythmically synthesised, and vice versa; however, the % of proteins that are both rhythmically synthesised and rhythmically abundant is 6 to 11--fold higher than would be expected by chance (taking proteins rhythmic by RAIN and ANOVA, respectively; in both cases the overlap between the two sets is highly significant) . This serves as a positive control, i.e., a minority of proteins show correlated rhythms of synthesis and abundance that are consistent with the canonical activity of 'clock-controlled genes' which cannot be explained by overestimation of rhythmicity.

```
Odds Ratio comparison synthesis vs total
Synthesis rhythmic by RAIN - listA size=148, e.g. A8Y5H7, B2RUR8,
E9Q4N7
Total rhythmic by RAIN - listB size=149, e.g. A1A5B6, A2A6T1, A2AI08
Intersection size=34, e.g. A8Y5H7, O08795, O54910
Union size=263, e.g. A8Y5H7, B2RUR8, E9Q4N7
Genome size=2528
# Contingency Table:
      notA inA
notB 2265 114
inB   115   34
Overlapping p-value=5.4e-13
Odds ratio=5.9
Overlap tested using Fisher's exact test (alternative=greater)
```

Jaccard Index=0.1

```
Synthesis rhythmic by ANOVA - listA size=66, e.g. A8Y5H7, O35639,
O55143
Total rhythmic by ANOVA - listB size=83, e.g. A8Y5H7, B2RQC6, E9Q6J5
Intersection size=16, e.g. A8Y5H7, P22561-2, Q3TB82
Union size=133, e.g. A8Y5H7, O35639, O55143
Genome size=2528
# Contingency Table:
      notA  inA
notB 2395   50
inB      67  16
Overlapping p-value=9.7e-11
Odds ratio=11.4
Overlap tested using Fisher's exact test (alternative=greater)
Jaccard Index=0.1
```

Nevertheless, we agree with the reviewer's general point and have revised the text as follows:

Page 9: *"... and may be susceptible to overestimation of rhythmicity."*

Page 9: *"Consistent with similar previous studies, <10% of detected proteins showed any significant variation over the circadian cycle (Fig. 2e)..."*

Page 9: *"The proportion of such proteins was more than expected by chance ($p < 0.0001$, Fisher's Exact Test), and their behaviour aligns with the canonical "clock-controlled gene" paradigm, in which physiological rhythms are proposed to arise through circadian variation in protein abundance, generated via transcriptional and translational oscillations."*

Methods, Page 21: *"...(n=1 per timepoint)"*

6) Why were the genes selected in 2C? these are not discussed anywhere else in the manuscript.

These are simply illustrative examples so that the reader can better understand what we mean, i.e., two proteins in different phases and one that did not change, all within a similar range of abundance. The selected proteins were not discussed because we do not expect the reader to attach any specific meaning to them. We have revised the figure to include in 2C examples of each rhythmicity category shown in 2E. To make this clear, we now state the following:

Figure 2 legend: *"No specific meaning is inferred from the protein identities"*.

7) The authors note that for Figure 2 "These observations are consistent with widespread rhythmic regulation of protein degradation." However, only 5-10% of the proteome is oscillating at any level and

less with a discrepancy between synthesis and abundance, so "widespread" is an exaggeration and this statement should be limited to the degradation in the rhythmic proteome.

We take the reviewer's point, but the term rhythmic proteome is also inaccurate since half the proteins with rhythmic degradation did not show an abundance rhythm in both mass spec experiments. We therefore revised this sentence as follows:

Page 10: *"These observations are consistent with widespread temporal organisation of protein degradation within the circadian-regulated proteome."*

8) The authors note that their more developed strategy in figure 3 would allow for the detection of less abundant proteins. However, they do not discuss that they in fact found less proteins overall, or if they were able to detect proteins of lower abundance. This is of some concern in determining if this is indeed the better method that they predict. How can the authors reconcile this issue? How can they rationalize this explains their increase in oscillating elements?

Thank you for raising this point, we did not explain ourselves sufficiently clearly. As stated in the revised text, once we had analysed the first iteration of pSILAC (Fig 2), we realised that detection of heavy-labelled proteins was *"inevitably limited and biased the proteome coverage towards abundant proteins with higher synthesis rates"*. In other words, in order to be considered in our analysis both unlabelled and heavy-labelled peptides needed to be detected in every sample at every time point. In fact, if we do not consider heavy-labelling, the overall coverage in the Fig 3 experiment (6577 proteins) was better than the Figure 2 experiment (6264 proteins), as expected, due to technical improvements in the methods used (by the time of the experiment in Fig. 3, we were able to perform the analysis using mass spectrometry techniques with better fractionation and detection, namely FAIMS and MS3). When the analysis criteria are applied however, this falls to 2302 and 2528 proteins, respectively. Because of the way that mass spectrometry works, many proteins needed to be excluded from analysis because the heavy label wasn't detected in one or more samples. In these cases, we cannot infer that no heavy-labelled protein was present in that sample or even that it was present at lower levels than other samples - it simply wasn't detected and therefore we cannot make any quantitative comparisons. Non-detection of any given heavy peptide may occur for several reasons, the most likely being that it co-elutes from the chromatography column at the same time as other much more abundant (light) peptides and simply escapes detection. This is an unavoidable limitation of the technique, we hope the reviewer can understand our need to restrict the analysis to those proteins whose nascent synthesis, and total abundance in the MMC fraction, can be confidently quantified.

As the experiments in Fig 2 and Fig 3 were performed independently, with separate TMT sets and different instrumentation, we are also unable to compare absolute abundances of the proteins between the two.

To communicate this more clearly we have amended Figures 2e and 3e to state the total coverage in the legends, as well as clearly stating the coverage of heavy-labelled proteins in the figure itself. We have also added the following explanation to the text:

Page 11:

“Despite enriching for only one cellular compartment, the overall coverage in this experiment was similar to the previous one (6577 and 6264 proteins, respectively), due to the altered and more targeted approach; with heavy peptides detected for 2302 proteins.”

9) In the comparison of complex turnover rates, the authors need to provide a metric that backs their statement that "the majority of component subunits not only showed similar average heavy to total protein ratios but also a similar change in synthesis over the daily cycle" for figure 3F.

Our apologies for this oversight, this is now presented in new Fig S3D.

10) In reference to the AHA incorporation, why is the hypothesis not that, like the puromycin, you would not see oscillation unless you add BTZ? Shouldn't the active degradation regulate the incorporation of AHA such that there is no visible rhythm unless you suppress degradation?

AHA is a methionine analogue that is sparsely incorporated into polypeptide chains with minimal effect on protein function/structure (Dietrich et al., PNAS, 2006). Unlike puromycin, therefore, AHA does not lead to chain termination or protein misfolding/degradation (Dermitt et al., Mol Biosyst, 2017) and so pulsed application at different phases of the circadian cycle is sufficient to reveal protein synthesis rhythms. The novelty in Fig 3H is the combination of AHA labelling with native PAGE that allows us to validate rhythmic production of high molecular weight protein complexes. This would not be possible with puromycin because prematurely-terminated polypeptide chains are not able to assemble into native complexes unless chain termination happens to occur at the extreme C-terminus and the C-terminus does not partake in any intermolecular interactions within the assembled complex.

11) The authors claim that there is enrichment of the actin cytoskeleton, but where this data can be found should be explained. The only thing that is shown is a few selected graphs of proteins in this pathway.

We previously reported circadian regulation of the actin cytoskeleton in Hoyle et al. (Sci Trans Med, 2017). The extremely high relative amplitude of Beta-actin (the structural component of microfilaments) in the MMC fraction is, in and of itself, entirely sufficient to demonstrate a circadian rhythm in the relative ratio of globular to filamentous actin that was originally identified by Ueli Schibler's lab (Gerber et al., Cell, 2013) and then shown to have a cell-autonomous basis in fibroblasts in Hoyle et al (2017). We have included further examples of an actin-binding protein (Corinin1b) and a motor protein (Myosin 6) to further illustrate this, but do not feel further discussion is warranted because it was comprehensively addressed in our previous work. The enrichment for actin was determined by GO analysis, which is now shown in the Fig 4A and referred to in the text.

The important point in Fig 4C is the difference in phase with the examples shown in Fig 4B and summarised in Figure 4A, i.e., there are a small number of proteins whose presence in the MMC fraction is highest in advance of the majority of rhythmically abundant proteins, but this earlier group doesn't show any significant synthesis rhythm. Actin is one of the most abundant cellular proteins, and by mass it

accounts for 67% of the circadian variation of rhythmically abundant proteins that peak in this fraction at the same phase. All these data and analyses are available for scrutiny in Supplementary Table 2.

To communicate this more clearly we have expanded on this point as follows:

Page 13: "*These proteins were enriched by 9-fold for actin and associated regulators of the actin cytoskeleton ($q < 0.05$, Fig. 4a and c). This is entirely consistent with circadian regulation of cytoskeletal dynamics and actin polymerisation that we and others have described previously (Hoyle et al, 2017, Gerber et al, 2013). Indeed as one of the most abundant cellular proteins, by mass alone, beta-actin accounted for 67% of the temporal compositional variation in the phase preceding ribosome biogenesis (Supplementary Table 2).*"

12) The authors note an oscillation in the total levels of p-eif2, commenting that these do not arise from the rhythms in total eif2a but temperature and feeding rhythms. However, unless I misunderstood, this work was done in fibroblast cell culture, so in this case, where would these temperature and feeding rhythms come from?

We were insufficiently clear. Daily rhythms of p-eIF2 have been observed under physiological conditions in mouse, *in vivo*. We do not observe similar rhythms in cultured fibroblasts under constant conditions unless the cells are challenged by stress. By inference therefore, it seems likely that daily rhythms of p-eIF2 *in vivo* arise from the interaction between cell-autonomous mechanisms and daily systemic cues such as, insulin/IGF-1 signalling and body temperature that are in turn driven by daily rhythms in CNS control, daily feed/fast rhythms and daily rest/activity rhythms, respectively. We have amended the text as follows:

Page 15: "*...and so suggest that daily p-eIF2 α rhythms in mouse tissues likely arise through the interaction between cell-autonomous mechanisms and daily cycles of systemic cues, e.g., insulin/IGF-1 signalling and body temperature rhythms driven by daily feed/fast and rest/activity cycles, respectively.*"

13) In Figure 5d, the treatment impeding degradation is causing cell death while the inhibition of translation does not. However, wouldn't too much, or not enough, translation, without compensatory regulation from degradation cause a problem in the same way that degradation does?

It is well-established that acute treatment with high concentrations of proteasomal inhibitors rapidly leads to proteotoxic stress that will trigger apoptosis unless resolved (Dantuma and Lindsten, *Cardiovasc Res*, 2010). Treatment with CHX is certainly stressful to cells, but in a different way, and cells die through mechanisms generally regarded to be necrotic and certainly do not involve the canonical proteotoxic stress responses that are activated by MG132 and similar drugs. Our findings show that, by whatever mechanisms cells die with CHX treatment, it does not change over the circadian cycle whereas death via proteotoxic stress does, consistent with our prediction. We hope the reviewer agrees it is beyond the scope of our study to explain why CHX-mediated cell death does not show a circadian rhythm in mouse fibroblasts.

Reviewer #1 (Significance (Required)):

The information that stems from this work is relevant and of interest to circadian clock field as how the regulation of the output of the circadian clock is implemented is still a major question in the field. This manuscript suggests a novel and plausible method for how, at least in part, this regulation occurs. However, the manuscript uses methods that do not measure degradation directly, which is a minor limitation. In addition, the mechanisms by which this regulation is imparted are not addressed in any meaningful way, even in the discussion.

We are sorry that we did not adequately discuss the extensive previous work that has already addressed regulatory mechanisms. We would like to stress that this manuscript concerns protein turnover and proteome renewal, of which degradation is obviously an important part but not the sole focus.

To communicate this more clearly, we have amended the title to:

"Circadian regulation of macromolecular complex turnover and proteome renewal"

... which we previously explicitly predicted in the discussion of previous papers (Feeney et al., Nature, 2016; O'Neill et al., Nat Comms, 2020; Wong et al., EMBO J, 2022) and our recent review (Stangherlin et al., Curr Opin Syst Biol, 2021).

With respect to measurement of degradation - Physiologically, cellular rates of proteasomal degradation are so intimately coupled with protein synthesis that, over circadian timescales, the former cannot meaningfully be studied in isolation. It is possible that the reviewer is alluding to historical methods that measure change over time in the presence of translational or proteasomal inhibitors, but these have long been known to introduce artifacts - because translational inhibition rapidly leads to reduced proteasome activity, whereas proteasomal inhibition rapidly reduces protein synthesis rates through the integrated stress response. We would be interested to hear of any more direct method for measuring protein degradation proteome-wide than the pulsed SILAC method we developed, as we are not aware of any. Even proteasomal proximity labelling coupled with MG132 treatment, recently developed by the Ori lab, does not directly measure degradation (bioarxiv <https://www.biorxiv.org/content/10.1101/2022.08.09.503299v1>). By definition, degradation can only be measured through the disappearance of something that was previously present, usually by comparing its rate of production with the change in steady state concentration (if any), which we have done using multiple methods.

With respect to regulation of degradation - We speculated on the mechanisms regulating rhythms in protein turnover in our several previous papers (Feeney et al., Nature, 2016; O'Neill et al., Nat Comms, 2020; Wong et al., EMBO J, 2021; Stangherlin et al, Nat Comms, 2021), whereas outside the circadian field these mechanisms have been addressed extensively. This was also discussed in detail in our recent review on the topic (see Stangherlin et al., COISB, 2021). In this review, we lay out the evidence for a model whereby most aspects of circadian cellular physiology might be explained by daily rhythms in the

activity of mammalian target-of-rapamycin complexes (mTORC). This model makes multiple predictions and informs the central hypothesis which is tested in the current manuscript: that circadian rhythms in complex turnover and proteome renewal should be prevalent over abundance rhythms. An enormous body of work over the last two decades has already clearly established mTORC1 as the master regulator of bulk protein synthesis and degradation, and a substantial number of independent observations have demonstrated circadian regulation of mTORC1 activity in vivo and in cultured cells. The mechanisms that drive cell-autonomous mTORC1 signalling are only partially understood (e.g. Feeney et al., Nature, 2016; Wu et al., Cell Metab, 2019), and we continue to explore this experimentally but they certainly lie well beyond the scope of this investigation.

Therefore, to address the reviewer's concern about inadequate discussion of mechanism, we have expanded on mTORC in the introduction and discussion, as follows:

Page 3: "Daily rhythms of PERIOD and mTORC activity facilitate daily rhythms of gene expression and protein synthesis. In particular, mTORC1 is a master regulator of bulk 5'-cap-dependent protein synthesis, degradation and ribosome biogenesis (Valvezan & Manning, 2019) whose activity is circadian-regulated in tissues and in cultured cells (Ramanathan et al, 2018; Feeney et al, 2016a; Stangherlin et al, 2021b; Mauvoisin et al, 2014; Jouffe et al, 2013; Sinturel et al, 2017; Cao, 2018). It is plausible that daily rhythms of mTORC activity underlie many aspects of daily physiology (Crosby et al, 2019; Stangherlin et al, 2021a; Beale et al, 2023b)."

Page 17: "The mechanistic underpinnings for cell-autonomous circadian regulation of the translation and degradation machineries remain to be fully explored, but are likely to be driven by daily rhythms in the activity of mTORC: a key regulator of protein synthesis and degradation as well as macromolecular crowding and sequestration (Stangherlin et al, 2021b, 2021a; Cao, 2018; Adegoke et al, 2019; Ben-Sahra & Manning, 2017; Delarue et al, 2018). In particular, global protein synthesis rates are greatest when mTORC1 activity is highest, in tissues and cultured cells, whereas pharmacological treatments that inhibit mTORC1 activity reduce daily variation in crowding and protein synthesis rates (Feeney et al, 2016a; Lipton et al, 2015; Stangherlin et al, 2021b). Given our focus on proteomic flux and translation-associated protein quality control, autophagy was not directly within the scope of this study but is also mTORC-regulated and subject to daily regulation (Ma et al, 2011; Ryzhikov et al, 2019). In vivo, daily regulation of mTORC activity arises primarily through growth factor signalling associated with daily feed/fast cycles (Crosby et al, 2019; Byles et al, 2021). The mechanisms facilitating cell-autonomous circadian mTORC activity rhythms are incompletely understood but may include Mg.ATP availability (Feeney et al, 2016a) and its direct regulation by PERIOD2 (Wu et al, 2019). This will be an important area for future work."

Reviewer #2 (Evidence, reproducibility and clarity (Required)):

Summary:

This is a very interesting and well written paper that addresses key questions in the circadian organization of proteostasis. The paper investigates origins of cellular circadian rhythms, invoking a premise early that there is a poor correlation between rhythmic gene expression - regulated by the canonical TTFL - and rhythms of the proteome, which are rather meager. Specifically, they ask how a relatively stable proteome

is possible if cells engage in rhythms of cellular protein synthesis? Their hypothesis is that protein degradation must rhythmically compensate for rhythms of synthesis and much of the manuscript is focused on defining the relationship between rhythmic global synthesis and rhythmic degradation. They employ a series of detailed proteomic investigations and biochemical assessments of protein synthesis coupled with various circadian reporters to assess proteasome function. The proteomic experiments reveal a limited number of proteins with oscillations in either synthesis or abundance or both and no discernible pathway organization however, a followup and more refined study that utilized fractionated samples and boosted heavy SILAC identified strikingly, that many proteins in relatively heavy fractions are rhythmic and that these fall into possible complexes including ribosome and chaperonins. Finally, they perform in vivo experiments testing whether the timing of proteotoxic stimuli regulates the degree of the integrated stress response measured as pEif2a.

Overall, I think that this is a fascinating paper that addresses an important question but falls short on mechanistically unifying them and completely contextualizing the findings in light of the canonical modes of circadian timekeeping leaving us with an important, but mostly descriptive set of findings. In addition, there are a number of important questions about data interpretation, some issues with data quality that should be addressed outlined below. With revision and further explication, this study will be an excellent addition to the growing field of circadian organization of the cellular proteome.

Thank you for reading and appreciating our work

Major and minor Comments.

Figure 1.

Fig 1a. The difference in Pulse and Chase at ZT24 does not appear to reflect the quantified data in 1b. This should be reconciled to make the figure convincing.

When working with radioactive cell lysates it is not possible to equalise the level of protein loaded on each gel beforehand as would happen with a western blot, for example. For this reason, the radioactive signal was normalised to the protein level subsequently measured by coomassie staining, as is standard practise for this type of assay, with all 4 replicates being shown in supplementary Fig.1A. An overnight phosphor screen image is presented in the main Fig.1A for illustrative purposes, but we take the point that this might not be immediately obvious. In revised Fig 1A we therefore now also show the relevant coomassie as well as labelling to make clear that the radioactive signal was normalised to protein levels.

How was the timing of the chase collection determined?

For these proof-of-principle experiments, we empirically determined the minimum duration of pulse and chase necessary to detect a quantifiable signal.

Fig 1d-e. What is the evidence that puro labeling results in 'rapid' turnover.

Apologies, this has been established for some time. Some additional papers are now cited in this section of the text (Liu et al, PNAS, 2012; Lacsina et al., PLoS One, 2011; Szeto et al., Autophagy, 2006)

Fig 1e seems to be missing the data from the treated and untreated conditions? How are the lines produced (e.g. linear versus rhythmic? Are these drawn lines or actual regressions?).

Fig 1e depicts the result of the experiment schematically explained in 1d. The only conditions were +Puro or +Puro+BTZ. There was no completely untreated condition, as puromycin incorporation is the basis of the assay (Lacsina et al., PLoS One, 2012; Szeto et al., Autophagy, 2006) and puromycin does not occur naturally in cells. We realise the figure could potentially be confusing without the associated raw data (anti-puromycin blots) - these are shown in supplementary Fig. 2A.

To explain the method more clearly, the following has been added to the results section where this experiment is described:

" As determined by anti-puromycin western blots, over two days under constant conditions, puromycin incorporation in the presence of BTZ showed significant circadian variation. In contrast, cells that were treated with puromycin alone showed no such variation, and nor did total cellular protein levels (Fig 1E, Fig S2A)."

The fit lines are produced by statistical comparison of fits, i.e., our hypothesis (damped cosine fit) vs null hypothesis (no or constant change over time, linear fit, $y = mx+c$), using sum-of-squares F test. The statistically preferred fit is plotted and p-value displayed on the graph, i.e., the regression line of the preferred fit and parameters are plotted. These details are reported in the figure legends.

Why was 30 minutes chosen as labeling time? It seems hard to understand here how protein degradation kinetics can be measured by puromycin labeling if the authors' claim that puromycin labeling potentially changes degradation rates as a function - primary or secondary - of the labeling itself. It seems they are measuring the potential to degrade proteins.

Puromycin labelling is a 20 year-old widely-used technique that can be employed in a range of applications. It was first used in a circadian context by Lipton et al (Cell, 2015) whose work we quickly followed (Feeney et al, Nature, 2016). Briefly, puromycin mimics tyrosyl-tRNA to block translation by labelling and releasing elongating polypeptide chains from translating ribosomes. When used at low concentrations (1 ug/mL in this case) puromycin is sparsely and sporadically incorporated into a small minority of elongating polypeptide chains. Those prematurely terminated chains have puromycin at the C-terminus, which can be detected by western blotting. We chose 30 minutes after optimisation experiments, as it was the shortest incubation time where a robust signal could be observed in these cells with this concentration of puromycin. The puromycinylated peptides are preferentially degraded by the ubiquitin-proteasome system because they are efficiently recognised as misfolded/aberrant proteins by chaperones within tens of minutes of being translated. Unless used at much higher concentrations, or over much longer timescales, there is no reason to believe that puromycin affects the degradation machinery itself, but the degradation of puromycinylated peptides depends on the proteasome. Therefore, puromycin+a proteasome inhibitor provides a reliable proxy for translation rate in the preceding 30 minutes, whereas puromycin alone tells us the steady state concentration under normal conditions, i.e., where proteasomes remain active. By subtracting the latter from the former we can infer the level of degradation of puromycinylated peptides that must have occurred in the previous 30 minutes. It is not a

perfect technique, but its results agree with other findings in this manuscript: that protein turnover varies more than steady state protein abundance. With respect to the potential to degrade proteins, this is measured in Fig 1C.

How do they determine that they are measuring degradation of functionally relevant proteins as opposed to a host of premature truncations?

We do not. This is measured by stable isotope labelling in Figures 2-4. Figure 1 provides the rationale for what follows in subsequent figures, i.e., proof-principle experiments suggesting that turnover is not constant over the circadian cycle. No single experiment in Figure 1 is expected to convince the reader that of circadian turnover. Rather, several independent methods suggest that bulk protein synthesis and degradation (turnover) are not constant over time, and deviate from the null hypothesis with variation that appears to change over the 24h circadian cycle.

Fig 1e bottom - again is this a true regression line?

It is not a regression line, otherwise a p-value of fit would be shown. Fig 1e bottom shows the bioluminescence measured at each timepoint from parallel control cultures (average of triplicates, error bars shown as dotted lines). Due to very high temporal resolution (every 30 min) and robustness of the cell line, it appears as a virtually perfect damped (co)sine wave. We apologise that this was not explained more clearly in the figure legend, now amended as follows:

"Parallel PER2::LUC bioluminescence recording from replicate cell cultures (mean +/- SEM, every 30 min) is shown below, acting as phase marker."

Perhaps two time points should be examined here - similar to the pulse chase performed with 35S labeling?

We are sorry we were not fully clear with our method here. The puromycin (+/- BTZ) labelling was performed over two days every 4h (so 12 timepoints in total), which can be inferred from the data points in the top two graphs in Fig. 1E, and x-axis - but is now also clearly stated in the figure legend. The bottom right graph was a continuous bioluminescence recording, integrated every 30 min from the set of parallel culture dishes. The bioluminescence data serves as a circadian phase marker, so that we can infer at which biological times synthesis and inferred turnover was higher vs lower.

We've adjusted the text to explain our method more clearly:

"Acute (30 min) puromycin treatment of cells in culture, with or without proteasomal inhibition (by bortezomib, BTZ), allowed us to measure both total nascent polypeptide production (+BTZ) and the amount of nascent polypeptides remaining when the UPS remained active (-BTZ). This allowed inference of the level of UPS-mediated degradation of puromycylated peptides within each time window, as a proxy for nascent protein turnover (Fig. 1D)."

Fig 1f. It appears that Puro labeling results in a rhythm between ZT1 and ZT13 but no statistic is provided and appears that the 'ns' is the results of variance in the data as opposed to difference in means? - would this not contradict the cellular result? What accounts for the rhythm reversal in the presence/absence of BTZ.

To be clear, we measured the level of puromycin incorporation in mouse liver *in vivo* following a similar method employed by Lipton et al, Cell, 2015 (Figure 2). The prediction was that, exactly as in cells (Fig 1E), treatment with a proteasome inhibitor would lead to a much greater increase in puromycinylated peptides at ZT13 than ZT1, because this is when protein synthesis is known to be higher and thus (we predict) protein degradation should also be higher. The experiment was not designed or powered to detect a time effect, it was designed to detect an interaction between time-of-puromycin treatment and BTZ, with the specific prediction being that BTZ would have a greater effect during the active phase. This is what we observed.

While the authors have previously demonstrated an increase in rhythmicity of the proteome in Cry1/Cry2 double knockout cells, it would have been welcome here to test a global loss of circadian transcription in the degradation assay. One might expect that these rhythms would also be even higher. What I am really asking is: what is the mechanism for rhythmic degradation and is it dependent on the canonical clock?

To address the reviewer's curiosity, we used the proteasome-Glo assay (also used in Fig 1C) to assess whether there was an interaction between genotype (WT vs CKO) and time at opposite phases of the circadian cycle over 2 days. We found a significant interaction by two-way ANOVA, indicating that components of the 'canonical clock' regulate the temporal organisation of proteasomal activity (see revised Figure S1). Circadian regulation of mammalian cellular functions, such as protein turnover, is a complex and dynamic process, whereas gene deletion affects the steady state and may be epistatic to phenotype rather than revealing gene function. We are therefore reluctant to speculate what this result means in the present manuscript, which is focused entirely on testing the hypothesis that global protein turnover and complex biogenesis have cell-intrinsic circadian rhythms in non-stressed, wild type cells.

To communicate this, the text has been revised as follows:

"Moreover, we detected a significant interaction between genotype and biological time when comparing trypsin-lik proteasome activity between wild type and Cryptochrome1/2-deficient cells, that lack canonical circadian transcriptional feedback repression (Fig S1B-E)."

Fig 2. How was the 'fixed window' timeframe determined?

A trial experiment was performed with labelling windows of various length, and 6h was determined to be the shortest window where enough heavy label incorporation was detected to be able to assess circadian changes. This was the case with our first methodology, which was subsequently improved (Figure 3), and therefore labelling window reduced to 1.5h.

Fig 3h. While admittedly difficult, the native PAGE is not of great quality and kind of unconvincing. Also not really sure why the AHA labeling is used here an nowhere else in the paper.

AHA is a methionine analogue that is sparsely incorporated into polypeptide chains with minimal effect on protein function/structure (Dietrich et al., PNAS, 2006). Unlike puromycin, therefore, AHA does not lead to chain termination or protein misfolding/degradation (Dermit et al., Mol Biosyst, 2017). In Figure 1, the aim was to validate previous reports of rhythmic protein synthesis assess whether there was any evidence for rhythmic turnover. To this end, we employed two independent methods (^{35}S -labelling and puromycin-incorporation). We did not want to rely on AHA for measuring turnover: although it has been validated and used for this purpose in some studies (McShane et al., Cell, 2016), AHA is not fully equivalent to methionine, and cellular aminoacyl-tRNA synthetases have much higher affinity to methionine than they do to AHA (Ma and Yates, Expert Rev Proteomics, 2018). It is thus impossible to perform AHA labelling without methionine-free medium, and in turn methionine starvation and media changes are known to have an effect on cell signalling and cell metabolism, which would be particularly pronounced in circadian context (over days rather than over hours).

By contrast, in Fig 3H, we use AHA with native PAGE to specifically validate one inference from the mass spectrometry analyses: circadian production of high molecular weight protein complexes. This would not be possible with puromycin because prematurely terminated polypeptide chains are not able to assemble into native complexes unless chain termination happens to occur at the extreme C-terminus and the C-terminus does not partake in any intermolecular interactions within the assembled complex.

The raw data (full gels, all replicates) are presented in Figure S2e, which of course was used for quantification. We have now picked a different example for the main figure, which hopefully allows for clearer representation.

The text in the results section describing the AHA experiment is now amended as follows:

" To validate these observations by an orthogonal method, we pulse-labelled cells with methionine analogue L-azidohomoalanine (Dieterich et al, 2006). AHA is an exogenous substrate, that cells have lower affinity to than methionine, and it could potentially impact on stability of the labelled proteins (Ma & Yates, 2018) – therefore, we only used AHA to assess nascent complex synthesis, rather than turnover. We analysed the incorporation of the newly synthesised, AHA labelled proteins into highest molecular weight protein species detected under native-PAGE conditions (Fig 3H, S3F). We observed a high amplitude daily rhythm of AHA labelling, indicating the rhythmic translation and assembly of nascent protein complexes. Taken together, these results show that daily rhythms in synthesis and degradation may be particularly pertinent for subunits of macromolecular protein complexes"

Fig 4. I was a little disappointed here that the authors did not directly assess macromolecular assembly of at least one of their "hits" and demonstrate functional relevance and most of the analysis is maintained at a very superficial, systemic level. STRING assemblies are not terribly helpful without clear k-means clustering or some other clearly visualizable metric for stratifying and organizing the putative PPI data - this figure (S3) could be markedly improved.

We agree that validation is important. The ribosome is by far the most abundant macromolecular complex in the cell, and was one of the major complexes to show clear evidence for circadian regulation of turnover, but not abundance, by our pSILAC proteomics. To validate this result, we took advantage of two important observations: (1) that all fully assembled ribosomes incorporate ribosomal RNA (rRNA) which can readily be separated from other cellular RNA by density gradient centrifugation; (2) pulse-labelling with heavy uridine-¹⁵N₂ allows nascent RNA to be distinguished from pre-existing RNA. Thus, combining stable isotope labelling with ribosome purification, we can distinguish nascently assembled ribosomes from total when the RNA is extracted, digested with RNase, and the % heavy/total UMP quantified by mass spectrometry. These data are presented in new figure 5, and are consistent with findings in Figures 3/4 that circadian regulation of ribosome turnover is prevalent over abundance, and that the phase of highest ribosome turnover coincides with the phases of high translation and turnover overall. We hope by addressing the reviewer's question by an entirely orthogonal method, they can share more confidence in our conclusions.

The statistical metric for STRING, specifically the p-value for enrichment in physical protein-protein interactions, is presented in the main Fig. 3G. It is now also reported in the legend for new Figure S4 itself.

Is it possible that some macromolecular complexes have rhythms because their constituent proteins have differential half-lives when in one complex compared with another in circadian time? This possibility was not discussed.

To our knowledge, there is no evidence that any major macromolecular complex in the cell has a functionally significant rhythm in abundance on a cell-autonomous basis. The reviewer's suggestion is an intriguing possibility, but we can think of no way that it could be measured, even in principle. The simplest interpretation of our data from the independent techniques we employ (pSILAC with fractionation, native PAGE + AHA incorporation) is a rhythm in synthesis.

Fig. 5.

Why is the first histogram in 3c not at unity?

This measures the average fold-induction in aggregation when cells are treated with MG132 for 4h at the indicated timepoints. Unity would indicate no induction at all, so the presented quantifications show that MG132 always elicited an increase in aggregation, with an effect size that varied with circadian phase.

Do ZT24 and ZT48 differ, similarly do ZT36 and ZT60?

No, neither difference is statistically significant (adjusted p-values of p=0.9 and p=0.07, respectively). This is now specified in the figure legend. Tendency to aggregate is also likely to change as a function of time in culture, which is why we think there is a slight increase overall in the second day of the experiment.

Fig S4f is not of good quality with missing eIF2a total and therefore no loading controls.

Thank you for prompting us to double-check this. We found that the levels of eIF2a were quite variable between the animals, and therefore we performed this experiment with 6 biological replicates. We have double-checked the quantification, and have now excluded 3 unreliable samples (the ones with undetectable levels of total eIF2a – ZT18 +BTZ replicate 1 & ZT18 -BTZ replicate 2, as well as ZT6 +BTZ replicate 4, where a smear does not allow for a reliable quantification of phospho-eIF2a) instead of 2 that were excluded originally. This still leaves at least 5 biological replicates in each group. In fact, the difference between BTZ and control in ZT6 is now deemed to be even more significant, going down to adjusted $p=0.0007$.

S4e? true regression lines?

The same method was used as in Figure 1. The fit lines are produced by statistical comparison of fits, i.e. our hypothesis (damped cosine fit) vs null hypothesis (no change over time, linear fit), using sum-of-squares F test. The statistically preferred fit is plotted and p-value displayed on the graph. These details are reported in the figure legends and methods section.

While I thought these experiments were effective, they did not tie back well to the rest of the paper. What are the consequences of a temporally sensitive ISR? Which pathways does it effect in circadian time? Here, the main holes in this study are somewhat exposed; namely, a lack of mechanistic depth in explaining the very fascinating, albeit mostly descriptive, findings. The implicit assumption made here is that aggregation is 'bad' but could the opposite be just as true? Taking these considerations in account would further strengthen the discussion.

The purpose of (former) Fig 5 was entirely to test the functional consequences and potential translational relevance of a daily rhythm in protein turnover. The mechanisms upstream and downstream of the ISR, and link with many diseases, are already quite well understood but we apologise that we did not draw more heavily on the prior literature to provide sufficient context for this experiment. Protein aggregation has long been associated with proteotoxic stress, and we do not assume it is good or bad, we simply use it as an additional validation of a temporally sensitive ISR. To correct this omission we have added the following to the results section before these experiments are introduced:

"Disruption of proteostasis and sensitivity to proteotoxic stress are strongly linked with a wide range of diseases (Wolff et al, 2014; Harper & Bennett, 2016; Labbadia & Morimoto, 2015; Hipp et al, 2019). Evidently, global protein translation, degradation and complex assembly are crucial processes for cellular proteostasis in general, so cyclic variation in these processes would be expected to have (patho)physiological consequences....

...Informed by our observations, we predicted that circadian rhythms of global protein turnover would have functional consequences for maintenance of proteostasis. Specifically, we expected that cells would be differentially sensitive to perturbation of proteostasis induced by proteasomal inhibition using small molecules such as MG132 and BTZ, depending on time-of-day."

Reviewer #2 (Significance (Required)):

This is a fascinating paper that addresses key questions in the circadian organization of the proteome. The paper's main findings are that rhythms of protein synthesis and degradation are temporally coordinated to maintain overall stability of the proteome in mouse fibroblasts. Furthermore, the authors present evidence that this temporal organization may be important for assembly of macromolecular complexes. While very interesting, the main limitations are a lack of biochemical and mechanistic explanation and evidence that verifies these, mostly descriptive, findings.

The fundamental biochemical mechanisms of protein synthesis, degradation, protein quality control and stress response have been studied for decades and are increasingly well understood, at least in cultured cancer cells. What is not understood is the extent to which all of these essential cellular systems are subject to physiological variation over the circadian cycle in quiescent cells. This is the fundamental knowledge gap our study attempts to fill by testing the discrete hypotheses that (1) circadian regulation of macromolecular complex turnover is more prevalent than abundance and that (2) proteome renewal is more prevalent than compositional variation. We suggest that establishing these essential principles of circadian cellular physiology is an essential prerequisite for performing the type perturbational experiments we presume the reviewer would prefer. We would like to reassure the reviewer that such studies have been and are being performed, but we are concerned that the inclusion of a very extensive additional body of work within this manuscript would detract from the clear communication of our major finding that complex turnover and proteome renewal has a cell-autonomous basis.

There are some relatively minor statistical and data quality issues that are probably addressable relatively quickly.

Upon revision the study would be a welcome addition to investigators interested in proteostasis, circadian biology, cell biology and proteomics.

I am a physician-scientist with expertise in circadian rhythms, cell biology, protein synthesis, and biochemistry.

Reviewer #3 (Evidence, reproducibility and clarity (Required)):

Seinkmane et al investigate circadian regulation of protein synthesis and degradation in cultured cells and in mice. Their main new finding is that protein synthesis and degradation are in many cases rhythmic but coordinated such that the proteome is rhythmically renewed without an apparent rhythm in total protein abundance. Particularly the pool of large protein complexes is rhythmically renewed in this fashion.

Using pulsed SILAC in combination with mass spectrometry, the authors are able to distinguish between total and newly synthesized protein levels in mouse lung fibroblasts. Analysis of these data shows that the synthesis of a large number of proteins is rhythmic although the total amount is constant, or that proteins are synthesized at a constant rate but the total amount is rhythmic, suggesting that degradation is rhythmic.

By analyzing macromolecular complexes, defined as a high-speed pellet, they also present evidence that

Full Revision

the rhythmic components of large complexes oscillate in the same phase and have a similar protein turnover rate. The authors conclude that complexes assemble rhythmically.

The authors also present evidence that the activity of the proteasome oscillates in a circadian manner. Based on this observation, they show (in fibroblasts and in mice) that the response to proteotoxic stress (monitored by eIF2alpha phosphorylation levels, protein aggregation, and apoptosis) is higher at circadian times of high proteasome activity.

I am an expert in the circadian field, and the hypothesis and concept behind the work presented here are potentially very interesting, and the experimental design is in principle suitable to answer these questions. However, after reading the paper several times, I cannot find the set of experiments that would convincingly support the authors' conclusions.

Major questions/points:

The major limitation of the manuscript is that the conclusions rely heavily on statistical analysis and massive processing of data from a bewilderingly large number of very different experiments. In looking at the figures, I have often wondered if the presence or absence of a rhythm is real or a product of the heavily processed data. The fact that a cosine wave fits through data points better than a straight line does not necessarily mean that a circadian rhythm is present.

We agree that comparison of fits alone does not provide sufficiently reliable evidence. However, the fact that many independent methods (cosinor, RAIN, ANOVA) yield similar overall findings lends more confidence to our findings. We would also argue that the large number of different experiments is a positive aspect of the paper and lends weight to the general conclusions. We instead ask the reviewer to consider an alternative question - we and many other labs have found no evidence for any change in total cellular protein content, and yet there is extensive evidence from independent labs for a 'translational rush hour' whilst (excepting some low abundance transcription factors) very few cellular proteins change by more than 10% over the circadian cycle (see Stangherlin et al, COISB, 2022 for extended discussion of this). We hypothesised a parsimonious explanation for this clear contradiction, and designed experiments whose data were analysed by widely used methods that yielded results that were consistent with prediction. Perhaps the reviewer will at least concede that, if the presented findings do not refute the hypothesis, it should not be rejected until a superior one is proposed?

I think that in particular, the SILAC experiment(s) should be repeated and also performed with an arrhythmic control (such as CRY1/2 KO).

Whilst we agree that CRY1/2 KO cells show no circadian regulation of transcription and much more variable rhythms in PER2::LUC activity than wild type controls (Putker et al., EMBO J, 2021), in our hands circadian rhythms in proteome composition and protein phosphorylation in CRY1/2 KO are at least as prevalent as in wild type cells (see Wong et al., EMBO J, 2022). Indeed, when we performed a proteasome activity assay in CRY1/2 KO fibroblasts, we observed there was an apparent circadian variation, similar to WT but with a different phase. These data are now presented in revised Figure S1.

Similarly, Lipton et al (Cell, 2015) showed circadian translational rhythms in cultured Bmal1 KO cells (see final figure), therefore it is not clear what would constitute an appropriate 'arrhythmic' control.

In this study, for proteomics experiments, we used a combination of SILAC and TMT, as each technique alone would not be sufficient to answer our specific questions. These two techniques are very resource-intensive on their own, and even more so in combination. We therefore had to prioritise and for the second SILAC-TMT experiment decided to focus on cellular fractionation and questions pertaining macromolecular complexes, which were directly relevant to our hypothesis. While it would undoubtedly also be interesting to study how canonical clock genes, such as *Cry1/2*, impact turnover on a proteome-wide scale, the focus of our study is physiological regulation of proteome composition, rather than the function of Cryptochrome genes which we already explored in previous work (Putker et al., EMBO J, 2021; Wong et al., EMBO J, 2022).

Comparability between the whole cell and MMC SILAC experiments is also limited due to the different experimental conditions (6h vs. 1.5h pulse, +booster).

We do not make any direct comparisons, other than to report that broadly comparable numbers of proteins were detected. Implicitly this means there must be greater coverage of protein complexes in the second pSILAC experiment, which our data bears out. If we were not to report the first experiment, the reader would not understand why we refined the method used in the second. In reporting the results of the 6h pulse, we make the limitations of this experiment very clear i.e. biased towards highly abundant, highly turnover proteins, irrespective of cellular compartment. We should add that even in this experiment there was a clear trend towards rhythmic turnover of ribosomal proteins, but this did not quite achieve significance ($p = 0.07$) and so we did not want to make claims beyond the data.

The essential and new message of the paper is that (at least some) macromolecular complexes undergo circadian renewal (degradation and synthesis). Rather than just analysing an operationally defined pellet fraction by mass spectrometry, this could be shown in more detail and directly for one or two specific macromolecular complexes. Ribosomes, for example, seem particularly suitable, because there would also be the very simple approach of measuring the synthesis of ribosomal RNA by pulse labelling. To me, such an analysis would be perfectly sufficient as a proof of principle. I would then omit aspects such as rhythmic stress response, since many additional experiments are needed to demonstrate this convincingly.

Thank you for the excellent suggestion, we agree that validation is important. The ribosome is by far the most abundant macromolecular complex in the cell and was one of the major complexes to show clear evidence for circadian regulation of turnover, but not abundance, by our pSILAC proteomics. To validate this result, we took advantage of two important observations: (1) that all fully assembled ribosomes incorporate ribosomal RNA (rRNA) which can readily be separated from other cellular RNA by density gradient centrifugation; (2) pulse-labelling with heavy uridine- $^{15}\text{N}_2$ allows nascent RNA to be distinguished from pre-existing RNA. Thus, combining stable isotope labelling with ribosome purification, we can distinguish nascently assembled ribosomes from total ribosomes when the RNA is extracted, digested with RNase, and the ratio of light to heavy UMP quantified by mass spectrometry. These data are presented in new figure 5, and are consistent with findings in Figures 3/4 that circadian

Full Revision

regulation of ribosome turnover is prevalent over abundance, and that the phase of highest ribosome turnover coincides with the phases of high translation and turnover overall. We hope by addressing the reviewer's question by an entirely orthogonal method, he/she can share more confidence in our conclusions.

The final figure is included because it tests predictions that were informed by the preceding experiments. It is not intended to be comprehensive exploration of how the integrated stress response changes with the circadian cycle, nor have we claimed this.

Specific points:

The reader is strongly influenced by the cosine wave or straight lines in the graphs (e.g. 1c, e, 3h, 5b, etc) produced by the analysis of rhythmicity, which basically only gives a yes or no answer. But it is not really that simple. If the algorithm detects a rhythm what is its period? Is it the same as the period of the luciferase reporter? If the period lengths correlate, do the phases as well (e.g. see differences in phases 1c and e)? These questions are not addressed.

The temporal resolution of the time course data is much lower than the luciferase reporter and so the error of the fit is greater (usually 1-2h). For the cosine wave curve fit and the associated extra sum-of-squares F test, the period of the oscillation was fixed at either 24h or 25h, as determined from a parallel PER2::LUC control recording. This is now explicitly stated in the methods section

In terms of phase, the general trend across all experiments is that bulk protein turnover, synthesis and degradation is higher during the 6-8h following the peak of PER2::LUC than at any other point in the circadian cycle. This is also consistent with our previous findings in mouse and human cells (Feeney et al, Nature, 2016; Stangherlin et al., Nat Comms, 2021) as well as findings from many different labs in vivo (e.g. Janich et al., Genome Res, 2016; Atger et al., 2015, PNAS; Sinturel et al., 2017, Cell). We are cautious about trying to be any more specific than this because each assay is measuring something different, and (as can be seen across the figures) there is also some modest variation in the phase of PER2::LUC between experiments, with respect the prior entraining temperature cycle (this will be reported in our forthcoming publication, Rzechorzek et al, in prep). To address the reviewer's point therefore, we have added the following to the discussion:

"Across all experiments in this study, we find that protein synthesis, degradation and turnover is highest during the 6-8h that follow maximal production of the clock protein PER2. This is coincident with increased glycolytic flux and respiration (Putker et al, 2018), increased macromolecular crowding in the cytoplasm, decreased intracellular K⁺ concentration and increased mTORC activity (Feeney et al, 2016a; Stangherlin et al, 2021b; Wong et al, 2022)."

The algorithm in Fig 1c predicts a rhythm for the chymotrypsin-like and the trypsin-like but not for the caspase-like activity. The peptide assay measures core proteasome activity independent of ubiquitylation and should therefore be dependent on proteasome concentration in the sample. How can then only two of the three proteasomal activities be rhythmic? Please elaborate and repeat with arrhythmic cells (e.g. CRY1/2 KO). The period length does not seem to correlate with the one of the reporter. Why is that?

The arrhythmic controls idea is partially addressed in the response above. We did perform a proteasome activity assay in CRY1/2 KO fibroblasts, and observed daily variation similar to WT, albeit with a different apparent phase. These data are now shown in Figure S1, and referred to in the main text as follows:

"Moreover, we detected a significant interaction between genotype and biological time when comparing trypsin-like proteasome activity between wild type and Cryptochrome1/2-deficient cells, that lack canonical circadian transcriptional feedback repression (Fig S1B-E)".

Besides this study, our two previous proteomic investigations of the fibroblast circadian proteome detected no biologically significant or consistent rhythm in proteasome subunit abundance (Wong et al., EMBO J, 2021; Hoyle et al., Science Translational Medicine, 2017). Moreover, proteasomes are long-lived stable complexes whose activity is determined by a combination of substrate-level, allosteric and post-translational regulatory mechanisms that includes their reversible sequestration into storage granules (Albert et al., PNAS, 2020; , Fu et al., PNAS, 2021; Yasuda et al., Nature, 2020). It is therefore very likely that the observed rhythm in trypsin- and chymotrypsin-like activity occurs post-translationally. Proteasome subunit composition is also known to change, which might be another reason for differences between the protease activities (Marshall and Vierstra, Front Mol Biosci, 2019; Zheng et al., J Neurochem, 2012).

To communicate this succinctly, we have revised the relevant text as follows:

Page 7: "Moreover, we detected a significant interaction between genotype and biological time when comparing trypsin-like proteasome activity between wild type and Cryptochrome1/2-deficient cells, that lack canonical circadian transcriptional feedback repression (Fig S1B, (Wong et al, 2022)). Previous proteomics studies under similar conditions have revealed minimal circadian variation in proteasome subunit abundance (Wong et al, 2022), suggesting that proteasome activity rhythmicity, and therefore rhythms in UPS-mediated protein degradation, are regulated post-translationally (Marshall & Vierstra, 2019; Hansen et al, 2021)."

Regarding period length, we apologise for an oversight in Fig 1c: unlike all other experiments presented here, these fits were originally done with a flexible period length (between 20h and 36h). This has now been re-fitted in a similar manner to the other experiments (fixed period of 24h, same as the parallel PER2::LUC controls), and the updated data are presented. This has not influenced the results of the statistical tests (only changed the p-values slightly, but the significance levels remain the same).

Fig. 1a,b suggest that there is a rhythm in global protein synthesis with a significant peak at 40h. Yet, Fig. 1e suggests otherwise. How can that be? Also, the degradation graph (lower panel 1c) has to be plotted with the ratios calculated from the data points and not the heavily processed fitted graphs. This can be very misleading.

Fig 1a,b was performed under quite different conditions to 1e. As described in the methods section, ³⁵S-labelling experiments require a medium change during both pulse and chase (to replace normal Met with radioactive Met, and vice versa). To avoid growth factor/mTORC1-mediated stimulation of protein synthesis & turnover, these acute media changes must occur in the absence of serum; otherwise media changes would introduce artifacts. In contrast, puromycin labelling (Fig 1e) is performed without any media changes (as puromycin can be added directly to culture cell media), and therefore was performed in normal culture conditions of 10% serum. Thus, due to its well-established effect of growth factor/mTORC1 signalling on bulk translation rate, it is very likely that differences in the phase of translational rhythms between Fig 1a,b and 1e are attributable to differing serum concentrations – this phenomenon of serum-dependency of phase is also described in Beale et al, 2023, bioRxiv <https://doi.org/10.1101/2023.06.22.546020>. The only important point, is that neither of these proof-of-principle experiments support the null hypothesis: that translation rate and turnover remains constant over the circadian cycle. Thus, the hypothesis being tested in Figure 1 is not rejected, and provides the rationale for the subsequent proteome-wide analyses.

With respect to 1E, given the variance of measurement, the curve fits to Puro and Puro+BTZ already serve to test whether there is any significant ~24h component, a ratio of the respective data points would simply compound the error of measurement. The degradation plot is provided purely for illustrative purposes to help the reader i.e. if these fits were true, what would be expected? We have revised the figure to more clearly communicate that the degradation plot is presented purely as a visual aid, labelled “inferred”, and now show ratio plots in revised Figure 1.

It also strikes me as odd that the amplitude of degradation increases (peak at 28h lower than at 30h) while the amplitude of the core clock oscillation dampens over time (peak at 54h higher than at 53h due to desynchronisation). Only two data values around 54h are responsible for the detected rhythm (2nd peak). Furthermore, phase and period do not agree with the rhythm of proteolytic activities shown in 1c. How can this be explained?

Due to the nature of the experiment, the degradation rate inferred from Figure 1B & 1E does not reflect proteasome activity exclusively. Rather it reflects the combined sum of processes that remove nascently produced proteins from the cell's digitonin-soluble fraction, which includes proteasomal degradation, but also autophagy, protein secretion and sequestration into other compartments. Therefore, the peak degradation in Fig 1B & E would not necessarily be expected to coincide with the peak of proteasome activity in Fig 1C. Again, these experiments in Figure 1 simply serve to test the hypothesis (change over circadian cycle) vs the null hypothesis (no change over the circadian cycle).

To the question of amplitude increase, we speculate that this is due to metabolic changes in cultures over the course of three days – as serum and nutrients from the last medium change at T0 are depleted, cells need to increase degradation to promote turnover and recycling. As we suggest that the rhythms in turnover help cellular bioenergetic efficiency, it is quite plausible that amplitude increases as nutrient-concentrations fall. We are in process of further investigation into how exactly these rhythms vary with nutrient and serum status.

Regarding the MS data shown in Figure 2, is it possible to show a positive / quality control? Best would

Full Revision

be MS data of Luciferase (or PER2,3, RevErb/alpha, DBP) to show oscillation of protein levels with the same phase and period as the reporter.

Unfortunately, none of these low abundance transcription factors were detected in our MS runs. This is not surprising, given that their copy numbers are estimated at <1000 per cell (OpenCell: Cho et al., Science, 2022). Since PER2::LUC is a fusion protein, LUCIFERASE is not expected to be any more abundant than PER2. However, in a previous study we measured circadian changes in LUC abundance directly from PER2::LUC cells (Putker et al., EMBO J, 2021), and found that it follows PER2::LUC activity in a manner consistent with the measured kinetics of firefly luciferase (Feeney et al., J Biol Rhythms, 2016).

In Fig. 2c examples of the 4 groups of proteins presented in 2e should be shown (both synthesis and total abundance arrhythmic, either one rhythmic or both rhythmic) and not just what appears to be random examples of rhythmic and arrhythmic proteins.

As also requested by another reviewer, we have revised the figure to include examples of each of the rhythmicity categories. No specific meaning is inferred from the chosen protein identities.

Is it possible at all to distinguish between synthesis/turnover and assembly/disassembly of macromolecular complexes in the MMC SILAC experiment? If so, how?

We followed the established protocol originally developed in our collaborator Kathryn Lilley's lab, where it has previously been shown that most proteins in the MMC fraction are in macromolecular assemblies (Geladaki et al, Nat Commun, 2019). Proteins that are rhythmically abundant in this fraction, but without an accompanying synthesis rhythm (e.g. Beta-actin, see Hoyle et al., Sci Trans Medicine, 2017) can be reliably assumed to arise solely from rhythmic assembly/disassembly i.e. they are captured in this fraction when assembled, but lost, and therefore not detected, in this fraction when disassembled. However, in the case of rhythmic synthesis and abundance, it is not possible with this technique to directly infer that rhythmic synthesis of a given protein is responsible for its rhythmic assembly in a complex, though they do correlate.

Therefore, our new figure 5 (with thanks again for this suggestion) approaches this by an orthogonal method, relying on the important observations that a) ribosomes incorporate ribosomal RNA (rRNA) b) this can be readily separated from most other cellular RNA by density gradient centrifugation and c) pulse-labelling with heavy uridine-¹⁵N₂ allows nascent RNA to be distinguished from pre-existing RNA. Using this technique, we validate a rhythm in production and assembly of mature ribosomes, with its peak consistent with the highest turnover time as measured in Figs 1 and 3, and MMC fraction proteomics (Supplemental table 3), at the descending phase of PER2::LUC.

Looking at Fig. 4b,c, what is the fraction of rhythmic proteins from the MMC experiment that also oscillate in either synthesis, total abundance or both in the whole cell? Is there a general correlation at all? Please show.

There were no correlations greater than would be expected by chance (the sets of proteins rhythmic in either synthesis or degradation did not overlap significantly between whole-cell and MMC fractions, as determined by an odds ratio test).

To communicate this we have added the following text:

"It is also worth noting that although there were small sets of proteins that were rhythmic in both whole-cell (Figure 2) and MMC fractions (Figure 3), in both synthesis and total abundance, none of these four overlaps were higher than would have been expected by chance."

Why is the phase of the oscillating proteins different in the two experiments (compare Figs. 2f,g and 4a) and does either of them match with the phase of the PER2::LUC reporter, which should be the peak synthesis phase of the clock?

This was a labelling error on our part, our apologies and thanks for drawing it to our attention. We had attempted to harmonise all these phase values so that they were mutually comparable between the two mass spec experiments, but omitted to update all the figures. They have now all been updated to be inter-consistent. From our experiments, the peak of PER2::LUC consistently precedes the timing of maximum bulk translation. This phase difference is, at least in part, attributable to the inactivation kinetics of firefly luciferase (see Feeney et al., J Biol Rhythms, 2016), i.e., under conditions of saturating luciferin substrate, PER2 protein abundance peaks several hours later than PER2::LUC activity when measured in longitudinal live cell assays.

Regarding the sensitivity to MG132 in Fig. 5b it doesn't make sense that, while eIF2alpha phosphorylation is arrhythmic in untreated cells and the levels of eIF2alpha phosphorylation are (apparently) not exhibiting a rhythmic change by administration of MG132 at different circadian timepoints, the ratio of P-eIF2alpha with and without MG132 suddenly is.

Please show in Fig. S4b quantifications of the individual experiments with and without MG132. What is presented in 5b is after all the ratio of ratios of quantifications of Western blots, each of which individually does not display any appreciable rhythm. For me this is two much of processing of data. In my opinion, the MG132 4h acute treatment must show a detectable rhythm.

We apologise for being unclear in this panel and description. Our hypothesis concerned the fold-induction of the p-eIF2alpha:eIF2alpha ratio changing as a function of MG132 and time. Our reasoning being that the ratio may be more biologically-relevant as it is the relative change that cells sense and respond to, and not the absolute abundance of p-eIF2alpha. We applied a quantitative, two-channel fluorescent antibody technique to enable detection and quantification of p-eIF2alpha and eIF2alpha from each replicate at each time point from the same band of the same blot. We agree that no p-eIF2alpha rhythm is evident from a cursory inspection of any of the blots. This is due to the innate variance between dishes in extracted protein concentration, as well as the levels of basal eIF2alpha and its phosphorylation, and is the reason that we took great pains to be as quantitative as possible using the two-channel immuno-detection (LICOR). Due to the natural and stochastic variation in eIF2alpha levels and extraction between replicates and over time, it is difficult to get identical eIF2alpha loading to reveal the overlying rhythm in p-eIF2alpha, and furthermore, identical loading would give a misleading impression of the level of temporal

Full Revision

variation of eIF2alpha levels. Quantification reveals temporal variation in the MG132 treated samples but not in the untreated controls (Supp Fig 5A) – suggesting that there may be circadian regulation of the cellular response to MG132 challenge, rather than a cell-autonomous p-eIF2alpha rhythm under basal conditions. We quantified fold-induction from MG132 vs untreated to present in Figure 6A. We have presented all the raw data in supplementary figure 5 for readers to validate through their own analysis.

Minor:

In Fig. 1f please show dot blot with error bars as well as the individual experiments in the supplementals. Please check the graph legend (N>=3?)

Thank you for pointing out these omissions. The dot blot with error bars is now shown in Fig. 1F, and the full gels are now included as Fig. S2B. The main figure legend for 1f has also had the following added (explaining the N numbers):

"Four mice were used per condition, but in some cases one of the four injections were not successful i.e. no puromycin labelling was observed and so no quantification could be performed (full data in Fig. S2B)."

Please explain the mechanism of the "booster" used in the second SILAC experiment.

The following has been revised in the text:

" Namely, we added a so-called booster channel: an additional fully heavy-labelled cell sample within a TMT mixture (Klann et al, 2020). When the mixture is analysed by MS, heavy peptides from the booster channel increase the overall signal of all identical heavy peptides at MS1 level; at MS2 and MS3 this results in improved detection of heavy proteins in the other TMT channels of interest, and is particularly advantageous for the proteins with lower turnover that would fall below the MS1 detection limit without the booster."

p10 3rd paragraph: S2e not S3e

Thank you, this has been fixed.

p12 last paragraph please add reference to Figs. 5f,g

Thank you, this has been added.

Reviewer #3 (Significance (Required)):

xxxxx

Dear John,

Congratulations on a great revision! Overall, the referees have been positive. However, there remain a few editorial items before we can move forward with acceptance that we ask you to attend to with a revised version (this will not go back to the referees for re-consideration).

When you submit your revised version, please also take care of the following editorial items and add this also to your point-by-point response.

1. Please provide an author checklist.
2. Please provide up to five keywords, which may or may not appear in the title, should be given in alphabetical order, below the abstract, each separated by a slash (/).
3. Please provide a data availability section after the materials and methods section. Please see our online guide for more information but in general this section would include omics data or original computer codes.
4. Please remove the author contribution section from the main manuscript.
5. Please review our new policy on conflict of interests on the EMBO author guide website and add this section, with the title: Disclosure and competing interests statement.
6. Please convert the file with the 5 supplementary figures into a PDF format and label "Appendix", please add to this Appendix a table of contents with page numbers and rename figures "Appendix Figure S1" etc. in the appendix file and update the names in the main manuscript accordingly.
7. We include a synopsis of the paper (see <http://emboj.embopress.org/>). Please provide me with a general summary statement and 3-5 bullet points that capture the key findings of the paper.
8. We also need a summary figure for the synopsis. The size should be 550 wide by 200-440 high (pixels). You can also use something from the figures if that is easier.
9. Figure callouts in the main manuscript must be in chronological order. Please update Figure 4 so that it is not called out before Fig 3H.
10. There are three supplementary tables that are missing. Please provide them either added to the appendix as "Appendix Table S1" etc. or as "Table EV1" in excel format, uploaded as one file per table and update the names in the main manuscript accordingly.
11. During our standard image integrity analysis we noted that replicates 2 and 3 may possibly be the same image in Supplementary Figure 2. Considering in your figure legend you mentioned that replicate 3 has a different order to 1 and 2, it appears to me that you've accidentally inserted replicate 2 twice (replicate 1 is clearly not the identical image). Please confirm with us whether this is the case or not and if so please update with the correct image.
12. In the figure legends: Please define the p values ****/**/* for figure 3d, and 6e-f.
13. Please indicate the statistical test used for data analysis in the legends of figures 2c, f-g; 3g; 4a; supplementary figures 3e; 4.
14. Please note that in figure 6b there is a mismatch between the annotated p values in the figure legend and the annotated p values in the figure file that should be corrected.
15. Please note that the box plots need to be defined in terms of minima, maxima, centre, bounds of box and whiskers, and percentile in the legends of supplementary figures 3c-d.
16. Please include N in the legends of figures 3f; 6b, e-f; supplementary figures 3a; 5c.
17. Although 'n' is provided, please describe the nature of entity for 'n' in the legends of figures 1c, e; 3h; 5b; 6a, c; supplementary figures 1b; 5d.
18. Please defined error bars for figures 1b, f; 3f, h; 5b; 6a-c, e-f; supplementary figures 3a; 5c-d.
19. Please include a scale bar and its definition for supplementary figure 5b.

I thank you for the opportunity to consider your work for publication and I look forward to reading your revision.

Warm wishes,
Kelly

Kelly M Anderson, PhD
Editor, The EMBO Journal
k.anderson@embojournal.org

Revision to The EMBO Journal should be submitted online within 90 days, unless an extension has been requested and approved by the editor; please click on the link below to submit the revision online before 7th Mar 2024:

Link Not Available

Referee #1:

The revision of Seinkmane et al., includes considerable clarifications and a swath of new quantifications, re-writes and an entirely new set of experiments. We were particularly happy with the new Figure, supporting rhythms in the macromolecular assembly of the ribosome. We are satisfied that our concerns have been addressed and think this is an exciting paper. We recommend acceptance.

Referee #2:

I am satisfied with this revision and feel that this manuscript would be of high interest to EMBO.

Rev_Com_number: RC-2022-01700

New_manu_number: EMBOJ-2023-116080

Corr_author: O'Neill

Title: Circadian regulation of macromolecular complex turnover and proteome renewal

The authors addressed the remaining editorial issues.

Dear John,

Congratulations on an excellent manuscript, I am pleased to inform you that your manuscript has been accepted for publication in The EMBO Journal. Thank you for your comprehensive response to the referee concerns and for providing detailed source data. It has been a pleasure to work with you to get this to the acceptance stage.

I will begin the final checks on your manuscript before submitting to the publisher next week. Once at the publisher, it will take about three weeks for your manuscript to be published online. As a reminder, the entire review process, including referee concerns and your point-by-point response, will be available to readers.

I will be in touch throughout the final editorial process until publication. In the meantime, I hope you find time to celebrate!

Warm wishes,
Kelly

Kelly M Anderson, PhD
Editor, The EMBO Journal
k.anderson@embojournal.org

Please note that you will be contacted by Springer Nature Author Services to complete licensing and payment information.
